# Retrieval of Aerosol Properties from Aerosol Optical Depth Measurements with High Temporal Resolution and Spectral Range

Angelos Karanikolas[1,2*], Benjamin Torres[4], Masahiro Momoi[3], Marcos Herreras Giralda[3], Natalia Kouremeti[1], Julian Gröbner[1], Lionel Doppler[5] and Stelios Kazadzis[1]

[1] World Optical Depth Research and Calibration Centre (WORCC), Physikalisch-Meteorologisches Observatorium Davos/World Radiation Center (PMOD/WRC), Davos Dorf, 7260, Switzerland
[2] Institute for Particle Physics and Astrophysics, ETH Zurich, Zurich, 8093, Switzerland
[3] GRASP SAS, Lille, 59800, France
[4] Laboratoire d'Optique Atmosphérique (LOA), University of Lille, Lille, 59000, France
[5] Deutscher Wetterdienst (DWD), Meteorologisches Observatorium Lindenberg (MOL-RAO), Lindenberg (Tauche), 15848, Germany

*Correspondence to*: Angelos Karanikolas (angelos.karanikolas@pmodwrc.ch)

**Abstract.** Several sun photometer networks worldwide include instruments for aerosol optical depth (AOD) observations, such as Global Atmospheric Atmosphere Watch-Precision Filter Radiometer (GAW-PFR) and Aerosol Robotic Network (AERONET). AERONET provides additional aerosol properties such as the detailed volume size distribution and the single scattering albedo through inversion modelling of sky radiance measurements. However, the data availability for such properties is limited due to the limited number of daily almucantar sky radiance scans and cloudiness. AOD is measured significantly more frequently as there can be one measurement even every minute. Also, the AOD measurements are affected only by clouds being too close or covering the solar disk. The Generalized Retrieval of Atmosphere and Surface Properties (GRASP) is a flexible inversion model to retrieve aerosol properties from various observations. One of its capabilities is the retrieval of the volume concentration, the volume median radius and geometric standard deviation for each aerosol size distribution mode and the separation of AOD to each mode using only spectral AOD as an input parameter (known as the GRASP-AOD application). Such properties are important for various applications, as the size of aerosols affects their interaction with solar radiation, clouds and radiative forcing modelling. Particle size also shows significant differences depending on the aerosol type such as dust or biomass burning. In this study, we selected four common stations of GAW-PFR and AERONET, used GRASP to retrieve the bimodal size distribution parameters from AOD measured by GAW-PFR instruments (PFRs) and validated the results for different conditions using AERONET data as reference. One of those sites includes a multi-year parallel timeseries from two different BTS spectroradiometers that combined can provide direct spectral irradiance (and as a result AOD) in the 300-2150 nm range. Using this dataset, we were able to investigate the effect and potential benefits of the increased spectral range on GRASP-AOD retrievals. This is mostly focused on the retrieval of the coarse mode volume median radius, which is particularly challenging with the filter radiometers measuring up to 862 or 1020 nm. We also assessed the performance for certain dust and biomass burning cases. Our results showed good agreement between PFR AOD-based and AERONET sky radiance inversions for AOD modal separation and volume concentrations. Significant improvement of the PFR-AERONET

intercomparison was also possible for the fine mode volume and effective radius when restricting the datasets to AOD at 500 nm > 0.1 and Ångström Exponent (AE) >1. Also, the results showed consistency with previous study regarding the validation of such retrievals using AERONET AOD. Focusing on conditions with high proportion of dust particles, we found consistent results with the general cases

Using AOD with a larger spectral range (from BTS spectroradiometer), we found that the wavelength selection may affect the results and that using longer wavelengths can increase the sensitivity of coarse mode volume median radius to AOD and improve the correlation of the GRASP BTS AOD-based and AERONET datasets. However, the available data were limited, so it is not clear under what conditions the inclusion of such wavelengths will result in more accurate retrievals or to what extent.

Finally, we were able to reproduce with GRASP the aerosol size characteristics of unusual biomass burning cases from the Canadian wildfires during 2023, but the results showed systematically increased fine mode radius and concentration compared to the AERONET output.

## 1 Introduction

Atmospheric aerosols are critical in atmospheric science and environmental studies. By scattering and absorbing solar radiation, they influence the amount of radiation that reaches Earth's surface, thereby impacting ecosystems' exposure to biologically active radiation (Horneck, 1995; Bais et al., 2018; Barnes et al., 2019), the efficiency of solar energy systems (Myers, 2005; Hou et al., 2022; Papachristopoulou et al., 2024), and the planet's energy balance (Hodnebrog et al., 2024). Over recent decades, aerosols have significantly contributed to variations in surface solar irradiance (Wild, 2012; Wild et al., 2021; Correa et al., 2024). They play a vital role in cloud formation and can modify cloud characteristics (Winkler and Wagner, 2022; Maloney et al., 2022).Improved aerosol monitoring is therefore an important factor to consider for reducing the uncertainty in the attribution of radiative forcing (Rosenfeld et al., 2014; IPCC, 2023) and improving weather forecasts (Glotfelty et al., 2019; Huang and Ding, 2021). Additionally, aerosols are significant air pollutants affecting human health, causing millions of deaths annually (Xiang et al., 2021; Yu et al., 2024).

Aerosol optical depth (AOD) is a key parameter in studying Earth's energy budget concerning aerosols (WMO, 2003). AOD quantifies the total extinction of solar radiation as it passes through the atmosphere due to aerosols.

AOD spectral dependence can be approximated by the Ångström law:

$$\tau_a = \beta\lambda^{-\alpha} \tag{1}$$

where β denotes the turbidity coefficient, λ is the wavelength, and α represents the Ångström exponent (AE). The turbidity coefficient β correlates with aerosol column concentration, while the wavelength dependence of τ, indicated by α, relates to aerosol size predominance.

AOD measurements are conducted using instruments that measure direct solar irradiance (DSI) under cloudless conditions at wavelengths minimally affected by gas absorption, reducing uncertainties in optical depth corrections for trace gases. Various sun photometer types are organized into global networks, including the Aerosol Robotic Network (AERONET) (Holben et al., 1998; Giles et al., 2019), the Global Atmosphere Watch-Precision Filter Radiometer (GAW-PFR) (Kazadzis et al., 2018b), and SKYNET (Nakajima et al., 2020). AERONET comprises over 500 stations worldwide, utilizing the CIMEL CE318-TS

sun and sky photometer (CIMEL) as its standard instrument (Barreto et al., 2016). GAW-PFR consists of 14 core and 14 associated stations globally, predominantly situated in remote areas. It employs the Precision Filter Radiometer (PFR) and incorporates the WMO reference instruments (PFR-Triad) for AOD measurements (Kazadzis et al., 2018b). SKYNET is composed of various instrument types divided into sub-networks, covering approximately 100 sites, primarily in East Asia and the western Mediterranean. Its primary instrument for AOD and aerosol property measurements is the PREDE-POM sun and

sky radiometer (POM) (Nakajima et al., 2020). Several studies have displayed good homogenization between the AOD of these two networks on short-term campaigns (Mazzola et al., 2012; Kazadzis et al., 2018a; Kazadzis et al., 2023) and long-term observations (Cuevas et al., 2019; Karanikolas et al., 2022). Other instruments, such as spectroradiometers can provide AOD observations with larger spectral range and resolution, although the accuracy can be limited by strong gas absorption at certain wavelengths (Kazadzis et al., 2007; Cachorro et al., 2009; Fountoulakis et al., 2019; Gröbner et al., 2023).

Aside from AOD, there are other aerosol properties that refer to the total aerosol column, such as the aerosol size distribution (SD) and the aerosol optical properties such as refractive index. The SD describes the volume concentration of aerosols in relation to their radius and can be typically approximated as a bimodal lognormal function (Schuster et al., 2006). SD can be described by six parameters (three for each mode, fine and coarse in our case): the fine and coarse mode components of the volume concentration ($C_{Vf}$ and $C_{Vc}$), the volume median radius ($R_{Vf}$ and $R_{Vc}$) and the geometric standard deviation ($\sigma_{Vf}$ and

$\sigma_{Vc}$) (Torres and Fuertes, 2021). The SD and additional aerosol properties (such as the real (RRI) and imaginary (IRI) part of the refractive index) are typically retrieved through the inverse modelling of sky radiance observations and AOD (Dubovik and King, 2000; Sinyuk et al., 2020). The main network providing such properties is AERONET. AERONET also provides the separation of AOD into each mode, fine mode AOD ($AOD_f$) and coarse mode AOD ($AOD_c$) through two different methodologies. One is the inversion of sky radiance (Dubovik and King, 2000) and the other is through the spectral

deconvolution algorithm (O'Neil et al., 2003).

The sky radiance scans are performed once per hour for solar zenith angles (SZA) < 54° and at four specific angles (eight scans per day) for SZA>=54° (Sinyuk et al., 2020), while AOD observations are typically performed with a temporal resolution in the range of 1 to 15 minutes depending on the instrument, time and location (Cuevas et al., 2019). However, there were methodologies developed to retrieve the SD parameters using only AOD observations (King, 1978; King, 1982; Wendisch and

von Hoyningen-Huene, 1994). Several newer studies include such methodologies (Schuster, 2006; Kazadzis et al., 2014; Perez-Ramirez et al., 2015; Torres et al., 2017). In Kazadzis et al., (2014), AOD is used to retrieve the total volume concentration ($C_{VT}$) and the effective radius ($R_{eff}$) through a linear estimation technique (Veselovskii et al., 2012). The Generalized Retrieval of Atmosphere and Surface Properties (GRASP) (Dubovik et al., 2014; Dubovik et al., 2021) is a flexible algorithm used for

retrievals of aerosol properties using observations from various instruments. It also includes the capability to retrieve the six SD parameters described earlier and the $AOD_f$ and $AOD_c$ as derived products using only AOD observations and prior knowledge or assumption of the aerosol refractive index (Torres et al., 2017). The methodology was validated for AOD from AERONET at different sites (Torres and Fuertes, 2021).

The size of aerosols plays an important role in several processes and applications. Size predominance affects the interaction of radiation with aerosols by altering their scattering (Witriol and Sindoni, 1992) and absorption capabilities (Tian et al., 2023). Large particles like dust show increased forward scattering (Liou, 2002; Cuevas et al., 2019; Liu et al., 2023), which affects the diffuse solar irradiance distribution and fraction (Li et al., 2023).

The size predominance can indicate the aerosol type and under certain conditions it is a main difference between anthropogenic and natural aerosols. Natural aerosols tend to be larger and contribute more to the coarse mode aerosols, with types such as dust (Mona et al., 2014; Barreto et al., 2022), pollen and other biogenic particles (except viruses) (Maser and Jaenicke, 1995; Mampage et al., 2022) and sea salt (Ackerman et al., 2023). Anthropogenic aerosol emissions are mostly in the fine mode (Xia et al., 2007; Deng et al., 2022). Finally, one of the main aerosol types on Earth is the smoke from biomass burning (mostly from large wildfires), which can be either natural or anthropogenic and corresponds mostly to the fine mode (Liou, 2002; Alonso-Blanco et al., 2014; Shi et al., 2019; Masoom et al., 2023). As aerosols are crucial for cloud nucleation, their size also plays an important role in water droplet and ice crystal formation hence in cloud cover and properties as well (Svenningsson et al., 1997; Levin et al., 2003; Hernández Pardo et al., 2019). The size of aerosols is also one of the main parameters affecting the transport range of aerosols and the deposition rate (Nicolae et al., 2019; Rodríguez-Arias et al., 2023), with larger particles showing reduced residence time in the atmosphere. Aerosol size is also related to health effects , with smaller aerosols being typically more dangerous (Kodros et al., 2018). The various effects of aerosol size distribution on solar radiation and health make it an important consideration in climate and air quality models (Gong et al., 2003) and an important source of uncertainty in radiative forcing calculation and attribution (Li et al., 2022; Zhang et al., 2024).

In this study, we used GRASP and AOD from PFR observations to retrieve the SD parameters. The aim is to assess the performance of such retrievals using only AOD at four wavelengths in the range of 368-862 nm. We also investigate the performance of GRASP retrievals under different conditions and aerosol types, as well as the effect of wavelength selection and spectral range through AOD retrieved from BTS spectroradiometer observations.

## 2 Instruments and methodology

### 2.1 Instrumentation and locations

To validate aerosol properties retrieved from PFR AOD (GRASP-AOD from PFR data hereafter), we chose four stations with several years of parallel CIMEL and PFR measurements. These stations also have different characteristics, so we could validate the retrievals under different conditions. The sites and time series are: Davos in Switzerland (2005-2022), Izaña in Tenerife, Spain (2004-2022), Hohenpeissenberg and Lindenberg in Germany (2013-2022).

Davos (46.8N, 9.8E) is a mountainous Alpine town in Central Europe with the station located at 1589 m above sea level (a.s.l.). Its atmosphere is generally pristine, with occasional intrusions of anthropogenic aerosols from the surrounding more densely populated areas and dust episodes from the Sahara Desert; hence, the seasonal patterns of AOD depend on the atmospheric circulation (Nyeki et al., 2012). The average AOD at 500 nm is below 0.1 (Nyeki et al., 2012; Karanikolas et al., 2022).

Izaña (28.3N, 16.5W) is a high-altitude site (2401 m a.s.l.) in the Canary Islands with a particularly clean atmosphere under background conditions, but there are several dust intrusions from the Sahara Desert leading to higher AODs. In Izaña, the AOD at 500 nm remains below 0.1 except during dust episodes that can lead to AOD > 0.5. Dust episodes are particularly frequent during July and August, when the number of days affected by them tend to exceed the number of days under background conditions (Barreto et al., 2022).

Hohenpeissenberg (47.8N, 11.0E) is a mountain station close to the Bavarian Alps at 989 m; hence, its characteristics are similar to those of Davos (low aerosol load, mostly fine particles), although its aerosol load is generally higher than that of Davos (Nyeki et al., 2012).

Finally, Lindenberg (52.2N, 14.1E) is a rural station in the region around Berlin in East Germany at an altitude of 120 m, so it is more affected by anthropogenic aerosols. It is expected to be more polluted than the other three stations and to include mostly fine particles (Doppler et al., 2024; Wacker et al., 2024).

### 2.1.1 PFR

The Precision Filter Radiometer (PFR), utilized by the GAW-PFR network (Wehrli, 2000), is designed to measure aerosol optical depth (AOD) and the Ångström Exponent (AE). The instrument conducts direct solar irradiance (DSI) measurements every minute across four nominal wavelengths: 368, 412, 500, and 862 nm. It is mounted on an independent tracking system to ensure continuous alignment with the Sun throughout the day. The device features a quartz window at its entrance, protecting internal components from external environmental conditions. The internal environment of the PFR is stabilized by filling it with dry nitrogen at a pressure of approximately 2 bar. The temperature is maintained at 20°C with a precision of ±0.1°C using a Peltier-controlled system. After sunlight passes through the quartz window, it is transmitted through interference filters, which isolate specific wavelengths with a bandwidth (full-width-at-half-maximum (FWHM)) of 3 to 5 nm before reaching a silicon photodiode detector. The instrument's field-of-view angle (FOV) at FWHM is approximately 2°. Measurements are performed as follows: every minute, the shutter opens for 10 seconds, during which 10 sequential measurements are taken at each wavelength. This setup minimizes filter degradation caused by prolonged exposure to solar radiation. Three PFRs in Davos (Switzerland) form the reference triad. Instruments at Mauna Loa (Hawaii) and Izaña (Tenerife) are calibrated every six months using the Langley Plot method (Toledano et al., 2018; Kazadzis et al., 2018b) and serve as stability checks for the reference triad also every six months. The first month after the stability check that the PFR returns to Mauna Loa or Izaña, it is calibrated again with the Langley Plot method. Instruments from other stations are calibrated in Davos against the reference triad. The Lindenberg PFR is calibrated every two years. The PFR in Hohenpeissenberg is calibrated every two or three years and the data is reprocessed based on the two calibrations initial and upon return to PMOD/WRC, Davos.

### 2.1.2 CIMEL

The CIMEL sun and sky photometer (Barreto et al. 2016; Giles et al., 2019), the primary instrument of the AERONET network, is used to measure AOD, AE, and a range of other columnar aerosol properties, including single scattering albedo (SSA) and size distribution (SD). The instrument is equipped with a two-axis robotic tracking system, enabling it to perform direct sun observations and sky radiance scans in multiple directions. The wavelengths measured vary by the instrument version, ranging from 340 nm to 1020 nm for some versions, while others extend up to 1640 nm. The maximum number of channels is 10. For

this study, CIMEL instruments with at least eight filters were used, measuring at 340, 380, 440, 500, 675, 870, 940, and 1020 nm. The 940 nm channel specifically observes water vapour content. Filter bandwidths (FWHM) are typically 10 nm, except for 340 nm, 380 nm, and 1640 nm (2, 4, and 25 nm, respectively). The measurement process involves a rotating filter wheel, which moves to select filters sequentially, completing a full cycle in approximately 10 seconds. This process is repeated twice more, yielding three consecutive measurements (triplets) within 1 minute. Triplet data are crucial for cloud screening (Smirnov

et al., 2000; Giles et al., 2019). A silicon detector records the radiation, while the instrument's 1.2° FOV to reject stray light during the sky radiance scans. To further enhance accuracy, a four-quadrant detector identifies the point of maximum solar intensity, ensuring the instrument points directly at the Sun. The instrument's schedule includes sky radiance scans at various scattering angles, which are used to retrieve aerosol properties at 440, 675, 870, and 1020 nm. AERONET provides public access to AOD data at three quality levels: Level 1.0 (unscreened), Level 1.5 (cloud-screened), and Level 2.0 (cloud-screened

with final calibration and quality assurance).

AERONET CIMELs from Izaña are calibrated on site every six months with calibration transfer against another CIMEL acting as reference. The two reference instruments are calibrated with the Langley plot method one at a time in rotation every three months. The calibration of the Davos CIMEL is performed at the Laboratoire d'Optique Atmosphérique - Université de Lille (LOA) by comparison with master instrument. Since 2018 a calibrated instrument is provided for exchanging the field

instrument on a yearly basis. Previously the instrument was shipped to LOA for calibration after 1-1.5 years of operation. Lindenberg and Hohenpeissenberg include two CIMELs that are transported to University of Valladolid alternatively one at a time, approximately once per year to be calibrated with calibration transfer from a master instrument.

### 2.1.3 BTS spectroradiometers

The BiTec Sensor (BTS) (Zuber et al., 2018; Zuber et al., 2021; Gröbner et al., 2023) consists of two array spectroradiometers,

each measuring the spectral DSI in different spectral regions. The FoV of the instrument is 3° FWHM with 2° plateau. The first covers wavelengths from the ultraviolet (UV) to near-infrared (IR) in the range of 300-1050 nm with a spectral resolution of 2.5 nm at full width half maximum (FWHM) and measures the irradiance with a silicon detector. The second spectroradiometer extends the range to the near-IR by measuring from 950 to 2150 nm with a resolution of 8 nm and uses an extended InGaAs detector. For each of the two spectroradiometers, a collimator ensures the measurement of DSI only and a diffuser is used as

the entrance optic. The spectroradiometers are mounted on a solar tracker to automatically follow the Sun. Both

spectroradiometers include temperature stabilization to avoid the effect of the environment on the instrument's performance. The instrument is calibrated to provide irradiance measurements in SI units ($W/m^2/nm$), which allows the retrieval of AOD using satellite-based top of the atmosphere solar irradiance (Gröbner et al., 2023). The calibration expanded uncertainty (at a 95% confidence coverage interval) decreases from 3% at 300 nm to 1.0% at 400 nm, remains at 1.0 % between 400 nm and

1400 nm and increases to 3% until 2150 nm. The AOD retrieval (Gröbner et al., 2023) includes corrections for the absorption of ozone ($O_3$). The wavelength channels at 1022.0, 1238.0, 1551.0, 2108.1 and 2129.8 nm were also corrected for the absorption of water vapour ($H_2O$), carbon dioxide ($CO_2$), methane ($CH_4$) and nitrous oxide ($N_2O$). For all wavelengths of the AOD above 1000 nm we use the measurements of the second BTS spectroradiometer. An overlapping region between 980 and 1020 nm is used as diagnosis for the compatibility of the two instruments to be combined in one dataset with large spectral

range. The ratio between them is usually well within 1%.

The BTS are calibrated yearly through measurements of irradiance from 1000W lamps in PMOD/WRC (Davos) optic laboratory. Every two months measurements of irradiance from portable 200 W lamps are used to monitor the stability of the BTS.

## 2.2 GRASP algorithm

GRASP (described in Dubovik et al., 2014 and Dubovik et al., 2021) is an inversion algorithm that uses the multi-term linear estimation techniques to retrieve aerosol properties from different types of observations (active and passive remote sensing instruments, both from ground-based and satellite instruments). In this study, we focus on the retrievals that require only AOD as input. AOD at more than one wavelength combined with an assumption of the refractive index, provides as the main output retrievals of the SD parameters ($C_{Vf}$, $C_{Vc}$, $R_{Vf}$, $R_{Vc}$, $\sigma_{Vf}$ and $\sigma_{Vc}$), and other derived products such as $AOD_f$ and $AOD_c$ or total

effective     radius     (Reff)     or     total     volume     concentration     ($C_{VT}$),     $AOD_f$     and     $AOD_c$ (https://aeronet.gsfc.nasa.gov/new_web/Documents/Inversion_products_for_V3.pdf , last access 23/12/2024).

GRASP requires a set of initial guesses for the parameters we intend to retrieve (in our case the SD parameters). The complex refractive index is not retrievable with only AOD as an input parameter, so it is required as an a-priori input parameter. GRASP includes a forward model to simulate the AOD observations using the SD parameters and refractive index, which can be run

separately (for example, to perform tests with synthetic data). During the inversion process, GRASP first uses the initial guesses of the aerosol properties to simulate the AOD and compare it with the AOD observations. Through an iterative process, it changes the combination of aerosol properties' values until it identifies the optimal solution through the maximum likelihood method.

## 2.3 Retrieval and validation methodology

To retrieve the SD parameters from AOD obtained with the PFR, we used the multi-initial guesses approach described in Torres et al., (2017) and Torres and Fuertes, (2021) for the GRASP settings. As proposed in those studies, we kept only data points corresponding to AOD > 0.02 at 440 nm. We also filtered the data according to the inversion residuals. A very high

inversion residual indicates that the forward model failed to reproduce accurately the observed AOD provided as input, and therefore the output size distribution does not fit well that input.

Inversions using only the four PFR wavelengths make numerical convergence easier compared with CIMEL and lower residuals may appear without indicating a better-quality retrieval. Therefore, we used a modified version of the criteria mentioned in the two aforementioned studies, to define a valid inversion. To retain an inversion, the absolute inversion fitting error must be below 0.01 if the AOD at 412 nm is below 0.5 and below $AOD_{412} \times 0.011 + 0.007$ if the AOD at 412 nm is above 0.5. The AOD absolute error at 500 nm has to be below $0.01 + 0.005 \times AOD_{500}$ These criteria are not particularly restrictive, as

they always include a threshold that does not exceed 0.01 (the typical AOD uncertainty at air mass 1). These criteria rejected 0.81% of the total GRASP-AOD inversions in Davos, 1.21% in Izaña, 4.25% in Hohenpeissenberg and 5.45% in Lindenberg, of those inversions that satisfied the criterion AOD > 0.02 at 440 nm. When comparing properties corresponding to the fine mode, we also kept only data corresponding to $AOD_f$ > 0.01 at 500 nm. For coarse mode properties, the threshold is $AOD_c$ > 0.01 at 500 nm.

To validate the GRASP-AOD from PFR data retrievals, we used the AERONET products as reference. For $AOD_f$ and $AOD_c$, we used as reference both sky radiance inversions (AERONET inversions from almucantar scans) and the output of the spectral deconvolution algorithm (AERONET-SDA). The other parameters are available only through AERONET inversions from almucantar scans. The comparisons between GRASP-AOD from PFR data and AERONET-SDA are point to point for coincident measurements with a maximum time difference of 30 seconds. On the other hand, the almucantar scans last

approximately 5 minutes. Therefore, for the GRASP-AOD from PFR data and AERONET inversions from almucantar scans comparisons, we used the median of all PFR measurements during a 5-minute period starting up to 30 seconds earlier or later from the almucantar scan starting time. Finally, to more clearly display the performance of GRASP-AOD, we include comparisons where we filtered the datasets according to their AOD differences (Sect. S2). The aim of this procedure is to minimize the effect of factors such as cloud contamination and imperfect instrument synchronization under high variability

conditions and better show the combined effect of uncertainties related to GRASP-AOD and the typical retrieval uncertainties of the AOD in the PFR observations (including factors such as uncertainties of calibration, gas absorption corrections, signal random noise and field of view). More details are available in the supplement section S1. These selection criteria retained 91.6% of the data points of PFR AOD, 88.5% of the AERONET inversions from almucantar scans and 90.8% of the AERONET-SDA.

The retrieval of the SD parameters using AOD requires the prior knowledge (or assumption) of the complex refractive index as input to GRASP. The refractive index affects the retrievals, especially through an anticorrelation between the real part and the radii or concentrations (Van de Hulst, 1957; Yamamoto and Tanaka,1969; King et., 1978; Torres et al., 2017). However, careful selection of the refractive index can reduce the retrieval error. In our case, since AERONET timeseries were available, we used as input the AERONET inversions from almucantar scans refractive index climatologies. However, such climatologies

are not available in most of the GAW-PFR stations. Therefore, it is important to investigate the effect that a refractive index assumption may have on the GRASP-AOD from PFR data retrievals. For this purpose, we selected a subset of the datasets

using only two years of data from Izaña and Lindenberg and one year from Davos and Hohenpeissenberg to repeat the GRASP-AOD from PFR data retrievals using only one value of refractive index for all sites and months. The fixed refractive index is 1.45 for the real part and 0.003 for the imaginary part. This value is selected as typical for continental European and urban sites (applicable for 3 out of 4 stations in our case), based on climatologies of Dubovik et al., 2002. Theoretically this would not apply to Izaña, but the Izaña climatology also did not show very large deviation. The AERONET inversions from almucantar scans climatologies we used at these stations vary in the range of 1.38-1.49 for the real part and 0.0005-0.0090 for the imaginary part (Sect. S3).

## 2.4 Methodology to investigate the spectral range effect

Torres and Fuertes (2021) and the previous parts of the study showed that the retrieval of $R_{Vc}$ is particularly challenging with GRASP-AOD applied to PFR and CIMEL standard version data. The PFR and CIMEL measure AOD over a limited spectral range at selected wavelengths. However, BTS spectroradiometers can provide a larger spectral range and resolution. One of our aims was to investigate the effect of using different wavelength selections of spectral AOD to retrieve the SD parameters using GRASP, especially for $R_{Vc}$. Taking advantage of the large range of BTS wavelengths, we selected sixteen wavelengths unaffected by strong gas absorption (so lower uncertainty of AOD retrieval) that increase the spectral range significantly compared to CIMEL, namely: 340.2, 368.1, 380.3. 412.1, 440.3, 500.7, 675.4, 747.4, 780.6, 862.9, 870.0, 1022.0, 1238.0, 1551.0, 2108.1 and 2129.8 nm. We also use seven of them (the closest to the CIMEL channels: 340.2, 380.3, 440.3, 500.7, 675.4, 870.0 and 1022.0 nm) to repeat the GRASP retrievals and compare with the output of all sixteen wavelengths. BTS AOD was available in Davos since September 2021 and we used data until September 2024.

To extrapolate the AOD at the selected wavelengths, we used the logarithmic form of Eq. 1 and a least squares linear fit on the observed BTS AOD to retrieve the AE and turbidity coefficient for each spectrum. The wavelengths used for the linear fit are: 340.2, 368.1, 380.3, 412.1, 440.3, 500.7, 675.4, 862.9, 870.0 and 1022.0 nm. The BTS dataset currently does not include operational cloud screening and quality assurance procedures, but the collocation with PFR gave us the capability to derive a useable data for the purpose of this study. We performed the data screening on three stages (fully described in Table 1). Firstly, we removed AOD and AE values that are too high or too low. Then we filtered the AOD according to the BTS-PFR comparison. Under conditions suspicious for cloud contamination, the selection criteria are stricter. However, the comparison with PFR is performed only at wavelengths present in the first spectroradiometer. To limit the erroneous data in the wavelengths above 1000 nm, we ensured that AOD at 400 nm is always larger than the AOD at 1551 nm and applied certain thresholds for the differences between observed and extrapolated AOD for all wavelengths above 1000 nm. These thresholds are selected based on the AOD levels, the typical AOD uncertainties and the statistics of those differences as shown in Fig. 6. The quality criteria for AOD retained 76.3% of the data points. The median AOD difference at 500 nm between BTS and PFR is 0.002 in both cases, while the standard deviation (St.dev.) reduced from 0.141 (all data) to 0.009 (selected dataset).

As the aim of this study is to investigate the effects of wavelength selection, we did not focus on optimising a combination of several GRASP settings per inversion for each wavelength selection; rather, used two sets of initial guesses for the

concentrations and radii per inversion, depending on AOD and AE and retained the inversion with the smallest residual. For all the other settings, we used fixed values. $R_{Vc}$ initial guesses were fixed to 1.75 μm. The settings were selected according to self-consistency tests (Torres et al., 2017). The settings and more details about the procedure are available in Sect. S4. We filtered GRASP-AOD from BTS data retrievals according to the inversion residual, as with the PFR (Sect. 2.3), but with some modification. Using more wavelengths results in larger residuals more easily. The same applies when using observed AOD in comparison to extrapolated AOD. Also, extrapolated AOD tends to show lower residuals compared to inversions from observed AOD. The criteria correspond to maximum values of the absolute inversion fitting error (abs-res) and the absolute error of AOD at 500 nm (abs-res_500). We present the thresholds for each case in Table 2. The inversion residual criteria rejected 0.3% of the data points corresponding to the final AOD selection. We also used AERONET inversions from almucantar scans level 1.5 data for the same period as reference. We also kept the AERONET inversions from almucantar scans data corresponding to an inversion sky residual <7% and a sun residual < 0.35% (optical residuals of the direct irradiance and sky radiance fitted by the model to the observations, described in https://aeronet.gsfc.nasa.gov/new_web/Documents/Inversion_products_for_V3.pdf).

To compare the GRASP-AOD from BTS data retrievals with AERONET inversions from almucantar scans, we used the median of all GRASP-AOD from BTS data measurements within the time period from one minute before the start of the almucantar scan to six minutes after the scan's starting time. From the comparisons, we kept only data corresponding to AOD > 0.02 at 440 nm. As for the case of GRASP-AOD from PFR data, we kept only data corresponding to $AOD_f > 0.01$ when comparing fine mode parameters and $AOD_c > 0.01$ for coarse mode parameters.

Table 1: The conditions that the AOD of BTS dataset satisfies to be included in the final selection for the investigation and validation of size distribution parameter retrievals. The data are filtered by the values of AOD, their comparison with the PFR dataset and in case of wavelengths longer than 1000 nm, by comparison with the extrapolated AOD through the Ångström approximation.

| Condition number | Parameter value | AOD difference BTS-PFR | AOD difference of BTS observed-extrapolated |
|---|---|---|---|
| Condition 1 | Retained only: <br> • 0<AOD<2.5 <br> • 0<AE<2.5 | ΔAOD<=0.1 (all common wavelengths) | - |
| Condition 2 | Applicable when: <br> • AOD>0.3 at 500nm <br> • AE<0.5 | ΔAOD: <br> • <=0.07 for 368.1 nm <br> • <=0.05 for 412.1 and 500.7 nm <br> • <=0.04 for 862.9 nm | - |
| Condition 3 | Retained only: <br> $AOD_{440} > AOD_{1551}$ | - | ΔAOD: <br> • <0.025 at 1022 nm <br> • <0.03 at 1238 nm |

- <0.035 1550 nm
- <0.05 at 2108.1 and 2129.8 nm

Table 2: The criteria to filter the GRASP-BTSGRASP-AOD from BTS data retrievals for each wavelength selection. Fewer wavelengths lead to easier model convergence, so we use stricter criteria. The criteria for 7 wavelengths in the 340-1020 nm range are identical to the criteria of Torres & Fuertes (2021).

| Number of wavelengths | Absolute total residual threshold | | Absolute residual at 500 nm threshold |
|---|---|---|---|
| | $AOD_{440} < 0.5$ | $AOD_{440} >= 0.5$ | All AOD values |
| 7 | 0.015 | $AOD_{440}$ x 0.016 + 0.007 | $AOD_{500}$ x 0.005 + 0.001 |
| 16 | 0.018 | $AOD_{440}$ x 0.019 + 0.009 | $AOD_{500}$ x 0.005 + 0.018 |

## 3 Results

### 3.1 GRASP - AOD retrievals from PFR data

In this section, we describe the results of the validation of GRASP-AOD from PFR data against AERONET inversions from almucantar scans and AERONET-SDA for the full time-series of the four stations and how the differences in the aerosol properties behave under different conditions.

### 3.1.1 Validation of GRASP - AOD inversions from PFR data

Here, we present the validation of retrievals of aerosol SD parameters by GRASP-AOD from PFR data. First, we show that the AOD comparison between PFR and CIMEL for all and the selected data. All median differences and standard deviations are within 0.01 with the exceptions of the standard deviations for all data only, at Lindenberg (both comparisons) and Davos (only for the comparison with AOD from AERONET inversions) (Tables 3 and 42). The comparisons between GRASP-AOD from PFR data and AERONET inversions from almucantar scans for $AOD_f$ and $AOD_c$ at 500 nm show excellent agreement as well (Fig. 1). For $C_{VT}$, we obtained larger relative differences and slope of the linear fit between the datasets, but the correlation remained excellent (R>0.95) (Fig. 2a). $R_{eff}$ also shows good correlation (R>0.8), but with larger variance and deviation of the slope from 1 (>1.5) (Fig. 2b). $AOD_f$ also shows similar results when compared to AERONET-SDA output (R=0.99 median difference -0.004 and standard deviation 0.009 across 210616 common selected measurements, Sect. S1). In Tables 5 and 6, we show the statistics of the comparisons for all parameters (median differences, standard deviations and correlation factors) for the entire dataset (Table 5) and the selected data (Table 6). More information is in Figs. S1-S3, where

we show the comparisons of $C_{Vf}$, $C_{Vc}$ between GRASP-AOD from PFR data and AERONET inversions from almucantar scans as well as the $AOD_f$ and $AOD_c$ comparisons with AERONET-SDA. In summary, the concentrations show larger correlation factors than the radii. However, $C_{Vc}$ shows the largest relative median difference and $R_{eff}$ the smallest. Also, $R_{eff}$ shows the largest relative standard deviation and $R_{Vf}$ the smallest. The seleced dataset shows similar results with the full dataset, with mildly lower standard deviations and higher correlation factors for most of the parameters ($C_{VT}$, $C_{Vf}$ and $R_{Vf}$ are exceptions).

The uncertainties of the inverted parameters vary depending on the conditions. AERONET provides point-to-point uncertainty for two of the SD parameters ($R_{Vf}$ and $R_{Vc}$). It also provides the root mean square error (RMSE) for $AOD_f$ and $AOD_c$ (O' Neill et al., 2003) corresponding to the AERONET-SDA retrievals. In the selected data, for $AOD_f$, 78.1% of the differences between GRASP-AOD from PFR data and AERONET-SDA are within the RMSE. For $AOD_c$, the same percentage is 82.4%. On the other hand, for $R_{Vf}$ the differences between GRASP-AOD from PFR data and AERONET inversions from almucantar scans within the uncertainties account for only 15.8% of the points and 9.5% for $R_{Vc}$.

Table 3: AOD differences at 500 nm between PFR and CIMEL (AERONET inversions from almucantar scans and AERONET direct sun AOD) for each one of selected stations for data without selection criteria. Both comparisons include only the selected data according to the criteria in section S2. P95th-P5th corresponds to the difference between the 95th and 5th percentiles.

| Location | Median difference | St.dev. | P95th-P5th | median AOD PFR | Number of measurements |
|---|---|---|---|---|---|
| **PFR - AERONET inversions from almucantar scans** | | | | | |
| Davos | -0.001 | 0.006 | 0.017 | 0.058 | 484 |
| Hohenpeissenberg | -0.007 | 0.006 | 0.019 | 0.088 | 1312 |
| Izaña | -0.001 | 0.005 | 0.014 | 0.033 | 1726 |
| Lindenberg | -0.001 | 0.018 | 0.046 | 0.121 | 1620 |
| **PFR – AERONET-DIRECT** | | | | | |
| Davos | -0.001 | 0.014 | 0.018 | 0.052 | 55636 |
| Hohenpeissenberg | -0.006 | 0.010 | 0.023 | 0.080 | 40613 |
| Izaña | -0.003 | 0.007 | 0.015 | 0.031 | 116822 |
| Lindenberg | -0.002 | 0.015 | 0.045 | 0.126 | 44953 |

Table 4: AOD differences at 500 nm between PFR and CIMEL (AERONET inversions from almucantar scans and AERONET direct sun AOD) for each one of selected stations. Both comparisons include only the selected data according to the criteria in section S2. P95th-P5th corresponds to the difference between the 95th and 5th percentiles.

| Location | Median difference | St.dev. | P95th-P5th | median AOD PFR | Number of measurements |
|---|---|---|---|---|---|
| **PFR - AERONET inversions from almucantar scans** | | | | | |

| | | | | | |
|---|---|---|---|---|---|
| Davos | -0.001 | 0.006 | 0.015 | 0.056 | 437 |
| Hohenpeissenberg | -0.007 | 0.005 | 0.017 | 0.082 | 1173 |
| Izaña | -0.001 | 0.004 | 0.013 | 0.033 | 1604 |
| Lindenberg | -0.001 | 0.007 | 0.025 | 0.116 | 1044 |
| **PFR – AERONET-DIRECT** | | | | | |
| Davos | -0.001 | 0.005 | 0.015 | 0.051 | 52135 |
| Hohenpeissenberg | -0.005 | 0.006 | 0.020 | 0.076 | 36957 |
| Izaña | -0.003 | 0.004 | 0.014 | 0.031 | 112673 |
| Lindenberg | -0.001 | 0.007 | 0.021 | 0.118 | 30786 |

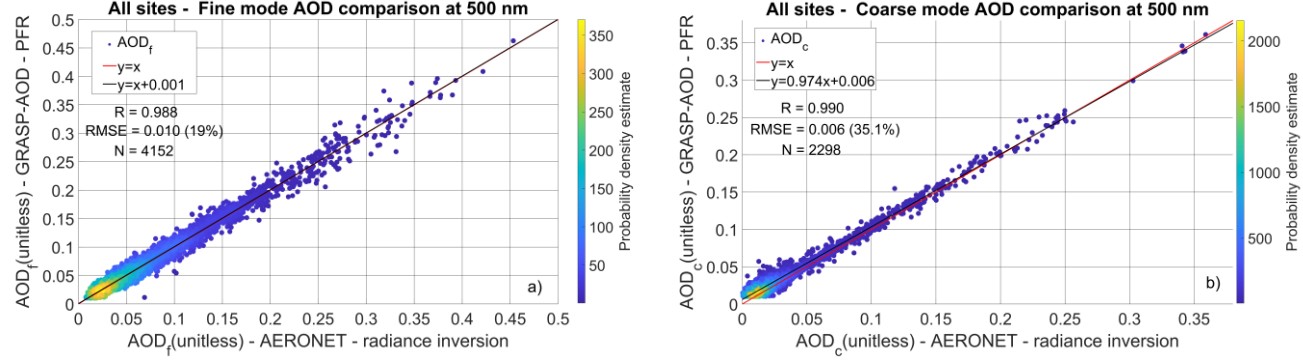

Figure 1: Scatter plot of $AOD_f$ (a) and $AOD_c$ (b) for the GRASP-AOD from PFR data and AERONET inversions from almucantar scans retrievals from all four locations. The plots include the correlation factor (R), the root mean square error (RMSE) and the number of observations (N). The colour bar shows the density of the points. We also include the linear fit between the datasets and the y=x line.

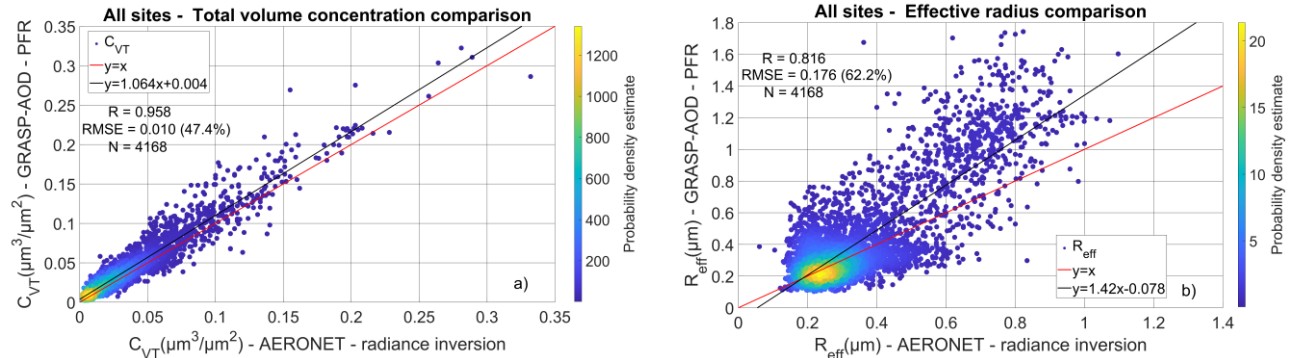

**Figure 2: Scatter plot of $C_{VT}$ (a) and $R_{eff}$ (b) for the GRASP-AOD from PFR data and AERONET inversions from almucantar scans retrievals from all four locations. The plots include the correlation factor (R), the root mean square error (RMSE) and the number of observations (N). The colour bar shows the density of the points. We also include the linear fit between the datasets (black line) and the y=x (red) line.**

Table 5: Statistics of the differences between GRASP-AOD from PFR data retrievals and AERONET inversions from almucantar scans for all data. We also include the AOD at 500 nm comparison between the PFR and AERONET inversions from almucantar scans. We also include the correlation factor (R) and the relative median difference compared to the median of each parameter from the reference dataset (PFR for AOD, AERONET inversions from almucantar scans for every other parameter).

| Parameter | Median difference | St.dev. | R | Relative median difference (%) | median of the parameter | Number of measurements |
|---|---|---|---|---|---|---|
| AOD 500 nm | -0.002 | 0.011 | 0.99 | -3.0 | 0.076 | 5400 |
| $AOD_f$ 500 nm | 0.000 | 0.013 | 0.98 | -0.1 | 0.055 | 5400 |
| $AOD_c$ 500 nm | -0.003 | 0.007 | 0.99 | -28.7 | 0.009 | 5400 |
| $C_{VT}$ | -0.005 | 0.012 | 0.96 | -21.5 | 0.022 | 5400 |
| $C_{Vf}$ | 0.000 | 0.006 | 0.89 | -4.7 | 0.009 | 5400 |
| $C_{Vc}$ | -0.004 | 0.011 | 0.97 | -41.5 | 0.010 | 5400 |
| $R_{eff}$ | -0.024 | 0.192 | 0.81 | -8.5 | 0.283 | 5400 |
| $R_{Vf}$ | 0.012 | 0.047 | 0.24 | 6.8 | 0.171 | 5400 |
| $R_{Vc}$ | -0.136 | 0.743 | 0.36 | -5.5 | 2.481 | 5400 |

Table 6: Statistics of the differences between GRASP-AOD from PFR data retrievals and AERONET inversions from almucantar scans for the filtered datasets. We also include the AOD at 500 nm comparison between the PFR and AERONET inversions from almucantar scans. We also include the correlation factor (R) and the relative median difference compared to the median of each parameter from the reference dataset (PFR for AOD, AERONET inversions from almucantar scans for
every other parameter).

| Parameter | Median difference | St.dev. | R | Relative median difference (%) | median of the parameter | Number of measurements |
|---|---|---|---|---|---|---|
| AOD 500 nm | -0.002 | 0.007 | 1.00 | -3.3 | 0.071 | 4168 |
| $AOD_f$ 500 nm | 0.000 | 0.010 | 0.99 | -0.8 | 0.050 | 4152 |
| $AOD_c$ 500 nm | -0.005 | 0.006 | 0.99 | -25.9 | 0.018 | 2298 |
| $C_{VT}$ | -0.004 | 0.010 | 0.96 | -20.6 | 0.021 | 4168 |
| $C_{Vf}$ | -0.001 | 0.005 | 0.89 | -6.6 | 0.008 | 4152 |
| $C_{Vc}$ | -0.008 | 0.011 | 0.96 | -39.9 | 0.019 | 2298 |
| $R_{eff}$ | -0.013 | 0.191 | 0.82 | -4.7 | 0.283 | 4168 |
| $R_{Vf}$ | 0.014 | 0.048 | 0.21 | 8.1 | 0.172 | 4152 |

| $R_{Vc}$ | -0.148 | 0.689 | 0.48 | -6.5 | 2.266 | 2298 |

### 3.1.2 Effect of refractive index on the GRASP-AOD inversions

In this section, we show the performance of the comparisons between GRASP-AOD from PFR data and AERONET inversions from almucantar scans retrievals for different refractive index selections (one fixed value in panel against climatology per site) (Table 7), including an example of scatter plots regarding $R_{eff}$ (Fig. 3). The comparisons between GRASP-AOD from PFR data and AERONET inversions from almucantar scans (for common datasets) tend to show better agreement when using the refractive index climatology, as expected, but the differences are very small (Table 7, Fig. 2). In Table 7, we summarize the results for all parameters (differences between the statistics of the GRASP-AOD from PFR data and AERONET inversions from almucantar scans comparisons for each refractive index selection). The results show that the effect of the refractive index is small for the refractive index selections we used in this study. Most differences of the median differences between GRASP-AOD from PFR data and AERONET inversions from almucantar scans in the two refractive index cases are close to 0. The differences in the correlation coefficients($\Delta R$) and the differences in the standard deviations ($\Delta$St.dev.) are also very small ($\Delta R$<=0.04). $\Delta$St.dev. is smaller than the standard deviation of the same parameter in Table 6 and typical uncertainty values of the four available parameters (Sect. S1).

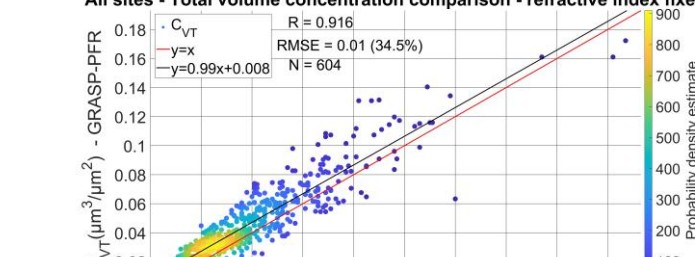
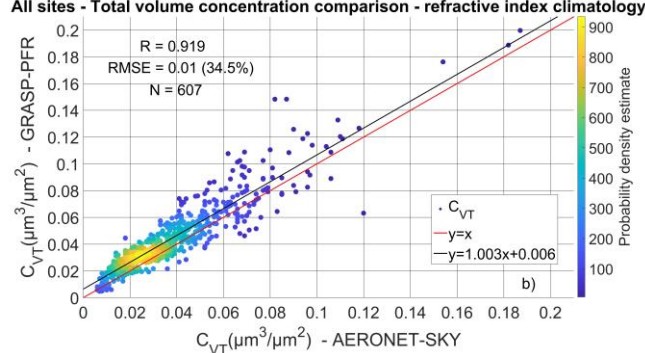

**Figure 3: Scatter plot of $C_{VT}$ for the GRASP-AOD from PFR data and AERONET inversions from almucantar scans retrievals from all four locations for a fixed value of refractive index (a) and the use of refractive index climatologies (b). The plots include the correlation factor (R), the root mean square error (RMSE) and the number of observations (N). The colour bar shows the density of the points. We also include the linear fit between the datasets and the y=x line.**

Table 7: Statistics of the differences between the GRASP-AOD from PFR data retrievals and AERONET inversions from almucantar scans comparisons for different refractive index selections, including R and the median of each parameter from AERONET inversions from almucantar scans retrievals.

| Parameter | Difference of the median difference | $\Delta$St.dev. | $\Delta$R | median of the parameter | Number of measurements |
|---|---|---|---|---|---|
| $AOD_f$ | 0.000 | -0.004 | -0.009 | 0.070 | 605 |
| $AOD_c$ | 0.000 | -0.001 | -0.008 | 0.042 | 159 |
| $C_{VT}$ | 0.001 | 0.000 | -0.011 | 0.029 | 604 |
| $C_{Vf}$ | 0.001 | 0.001 | -0.003 | 0.011 | 605 |
| $C_{Vc}$ | 0.001 | 0.001 | 0.011 | 0.036 | 159 |
| $R_{eff}$ | 0.002 | 0.000 | -0.014 | 0.319 | 604 |
| $R_{Vf}$ | 0.004 | -0.001 | -0.001 | 0.168 | 605 |
| $R_{Vc}$ | -0.019 | 0.016 | -0.037 | 1.835 | 159 |

### 3.1.3 Sensitivity of the retrieval of aerosol properties to the aerosol conditions

In general, aerosol properties inversions tend to be more accurate at higher AODs (Sinyuk et al., 2020). In Sects. 3.1.1 and 3.1.2, we showed that GRASP-AOD from PFR data performed well for the AOD modal separation and concentrations even under particularly low AOD conditions (AOD at 500 nm below 0.05). In this section, we investigate the performance of the radii in relation to the AOD and AE values and identify conditions under which the comparisons with AERONET inversions significantly improve.

As aerosol load and size affect AOD and AE, we could expect that the retrieval of aerosol properties should improve at higher AOD, as well as higher AE for small aerosols (fine mode) and lower AE for larger aerosols (coarse mode). In Fig. 4, we show that the $R_{eff}$ differences increase at very low AODs, but mostly at low AEs (particularly below 1, where we observe a positive bias towards larger GRASP-AOD from PFR data values). The same phenomenon is evident for $R_{Vf}$ as well (Fig. 5a and 5b). By further restricting the dataset (AOD at 500 nm > 0.1, $AOD_f$ > 0.04 and AE > 1), we achieved a significant improvement in $R_{Vf}$ in terms of correlation and RMSE of the linear fit between GRASP-AOD from PFR data and AERONET inversions from almucantar scans (scatter plots in Fig. 5c and 5d). There was also improvement for $R_{eff}$, but not for $R_{Vc}$ (Table S1).

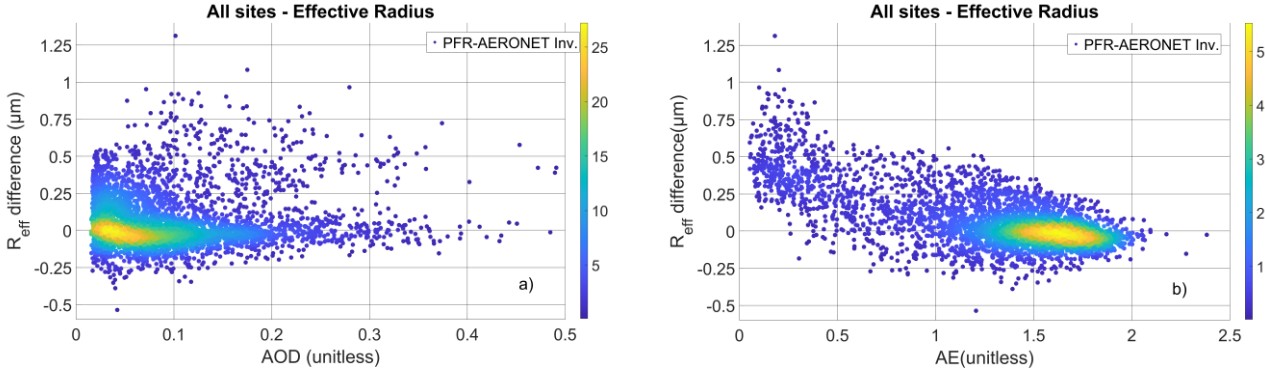

**Figure 4: The $R_{eff}$ difference between GRASP-AOD from PFR data and AERONET inversions from almucantar scans in relation to AOD (a) and AE (b).**

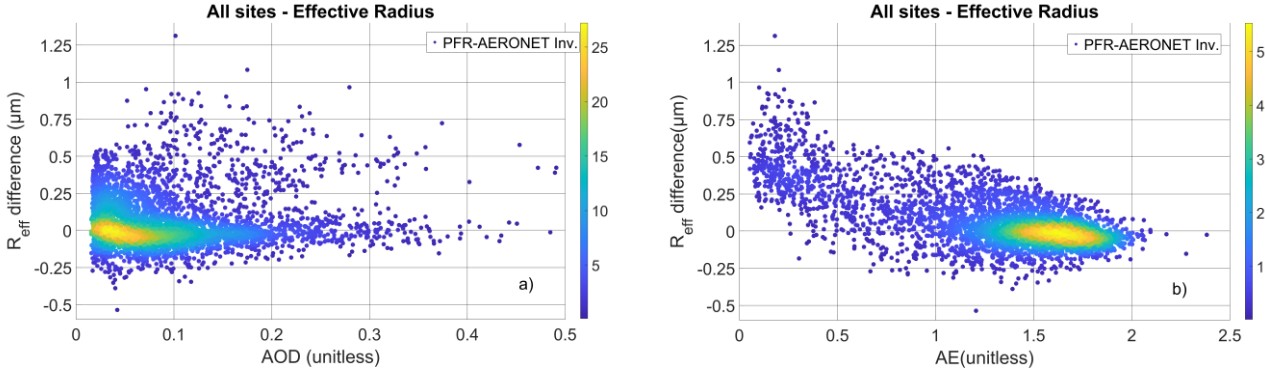

**Figure 5: The $R_{Vf}$ difference between GRASP-AOD from PFR data and AERONET inversions from almucantar scans in relation to AOD (a) and AE (b). Also, scatter plots for $R_{Vf}$ from GRASP-AOD from PFR data and AERONET inversions (c and d) under different thresholds of AOD and AE in data screening. In panel d) we have increased the AOD, $AOD_f$ and AE thresholds, which improved the comparison. All graphs correspond to all four locations.**

**3.2 Effect of AOD spectral range**

In this section, we present the results of the sensitivity study to the wavelength selection for GRASP-AOD from BTS data retrievals. In Fig. 6, we show the deviations of observed BTS AOD (AOD-obs) from the interpolated or extrapolated AOD using Eq. 1 (AOD-ext). The median AOD differences are <0.011 for all wavelengths. However, there are cases (especially in the UV or IR) where the 5th, 80th and 95th percentiles exceed 0.02, either due to noise in the observed AOD or because aerosol conditions cause the AOD spectral dependence to deviate significantly from the Ångström law, as it has happened with certain smoke cases (Eck et al., 2023; Masoom et al., 2025).

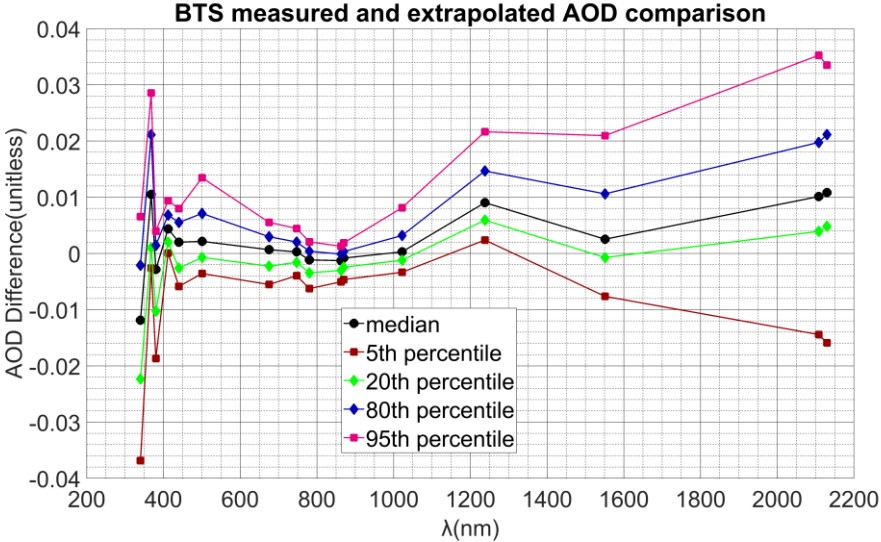

**Figure 6: The statistics of the differences between the BTS observed AOD and BTS AOD extrapolated using the Ångström law.**

As we know from Torres and Fuertes, (2021) and as we show in Sects. 3.1 and 3.3, $R_{Vc}$ showed low sensitivity to AOD. However, the coarse mode is generally more sensitive to longer wavelengths. Indeed, using the selection of sixteen wavelengths that cover the BTS spectral range, we see that the distribution of $R_{Vc}$ output of GRASP-AOD from BTS data shows larger variance compared to the seven wavelengths in the range 340-1022 nm (Fig. 7). However, the median $R_{Vc}$ shows only a small difference between the wavelength selections.

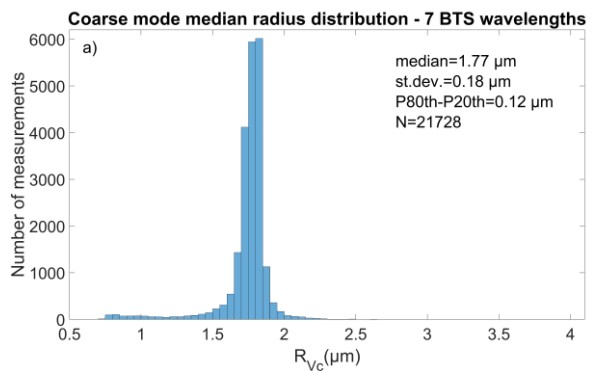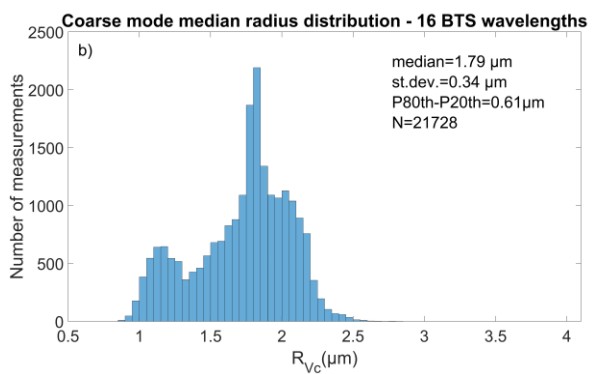

**Figure 7: Histograms of the $R_{Vc}$ GRASP-AOD from BTS data retrievals using AOD at 7 wavelengths (a) and 16 wavelengths (b).**

As shown in Table 8 (statistics of the differences between the GRASP-AOD from BTS data retrievals), the two AOD datasets produced different GRASP-AOD from BTS data output for all SD parameters. The median differences vary by parameter and we see no wavelength selection showing consistently larger or lower biases compared to the other. All parameters except $R_{Vc}$ show good correlation (R>=0.8).

The comparison between the GRASP-AOD from BTS data datasets with AERONET inversions from almucantar scans includes a particularly small number of measurements due to low data availability of AERONET inversions from almucantar scans data and low AOD in Davos, particularly in the coarse mode. The differences between GRASP-AOD from BTS data and AERONET inversions from almucantar scans are not consistently smaller or larger for the same wavelength selection (statistics shown in Tables 9-10). We repeated the coarse mode properties' and $R_{eff}$ comparisons for $AOD_c$>0.02 and FMF<0.8 at 500 nm (Table 10), because the presence of coarse mode aerosols in Davos is limited. For $AOD_{c/f}$ and the concentrations the results are similar. For $R_{eff}$ and $R_{Vf}$, using sixteen wavelengths showed decreased the biases and standard deviations, but also decreased the correlation factor, although the differences are not very large. For $R_{Vc}$, the correlation factor substantially increased when using sixteen wavelengths both for the full dataset (Table 9) and the dataset focused to the coarse mode (Table 10). The standard deviations also decreased, but the median differences increased.

The GRASP-AOD from BTS data comparisons with AERONET inversions from almucantar scans showed good consistency with the findings of the GRASP-AOD from PFR data and AERONET inversions from almucantar scans comparison, despite the different instruments and datasets. Comparing the median differences, standard deviation and R in Tables 9 and 10 to the corresponding from Table 6, we find that:

The correlation factors for $C_{Vf}$, and $R_{Vf}$ and $R_{Vc}$ differ by more than 0.1 compared to the ones in Tables 5-6 (GRASP-AOD from PFR data against AERONET), but $C_{Vf}$ and $R_{Vf}$ show smaller difference between the wavelength selections.

For $AOD_f$ and $AOD_c$, all absolute median and standard deviation of differences, differ by no more than 0.01 between the comparisons in Tables 6 and 9.

For the remaining parameters, the absolute relative median and standard deviation of the differences, differ by 0% and 69.5%, depending on the parameter and BTS AOD dataset. For $AOD_f$, $AOD_c$, $C_{VT}$ and $R_{Vc}$ using AOD at seven wavelengths shows

better consistency of median differences with the comparisons in Table 6 compared to the sixteen-wavelength selection. The same is true for the standard deviation except for the comparisons of $C_{Vf}$ and $R_{Vf}$. AOD at seven wavelengths also yielded the smallest $\Delta R$ only for $R_{eff}$ and $R_{Vf}$. The larger correlation of PFR with AERONET in RVc maybe related to the use of three initial guess instead of one, which created a trimodal distribution around them, while in the case of seven-wavelength selection of BTS it is unimodal.

Table 8: Statistics of the differences between the GRASP-AOD retrievals from BTS data when using 16 wavelengths minus when using 7 wavelengths. The first row corresponds to AOD at 500 nm simulated by GRASP for each one of the two wavelength selections.

| Parameter | median difference | St.dev. | P95th-P5th | R | Number of measurements |
|---|---|---|---|---|---|
| AOD fitted | 0.000 | 0.001 | 0.003 | 1.00 | 21728 |
| $AOD_f$ | -0.005 | 0.008 | 0.019 | 0.98 | 21728 |
| $AOD_c$ | 0.005 | 0.008 | 0.018 | 0.99 | 21728 |
| $C_{VT}$ | 0.005 | 0.017 | 0.048 | 0.95 | 21728 |
| $C_{Vf}$ | 0.000 | 0.002 | 0.005 | 0.94 | 21728 |
| $C_{Vc}$ | 0.006 | 0.017 | 0.047 | 0.95 | 21728 |
| $R_{eff}$ | 0.047 | 0.198 | 0.661 | 0.80 | 21728 |
| $R_{Vf}$ | -0.026 | 0.032 | 0.100 | 0.81 | 21728 |
| $R_{Vc}$ | 0.029 | 0.364 | 1.189 | 0.10 | 21728 |

Table 9: Statistics of the differences between the GRASP-AOD from BTS data retrievals and the AERONET inversions from almucantar scans for sixteen wavelengths in the 340-2130 nm range and for seven wavelengths in the 340-1022 nm range. The first row of data shows the comparison of the AOD at 500 nm corresponding to the AOD BTS observations and the almucantar scan inversions.

| Parameter | 16 wavelengths | | | 7 wavelengths | | | Number of measurements |
|---|---|---|---|---|---|---|---|
| | median difference | St.dev. | R | median difference | St.dev. | R | |
| AOD obs. | 0.003 | 0.009 | 0.99 | 0.003 | 0.009 | 0.99 | 63 |
| $AOD_f$ | -0.010 | 0.010 | 0.87 | -0.002 | 0.012 | 0.84 | 63 |
| $AOD_c$ | 0.010 | 0.011 | 0.99 | 0.003 | 0.010 | 0.99 | 63 |
| $C_{VT}$ | 0.002 | 0.022 | 0.96 | 0.003 | 0.018 | 0.94 | 63 |

| | | | | | | | |
|---|---|---|---|---|---|---|---|
| $C_{Vf}$ | -0.003 | 0.005 | 0.59 | -0.003 | 0.005 | 0.59 | 63 |
| $C_{Vc}$ | 0.007 | 0.020 | 0.96 | 0.004 | 0.017 | 0.94 | 63 |
| $R_{eff}$ | 0.139 | 0.136 | 0.78 | 0.210 | 0.240 | 0.85 | 63 |
| $R_{Vf}$ | 0.008 | 0.051 | 0.15 | 0.030 | 0.067 | 0.29 | 63 |
| $R_{Vc}$ | -0.469 | 0.428 | 0.73 | -0.387 | 0.626 | -0.04 | 63 |

Table 10: Statistics of the differences between the GRASP-AOD from BTS data retrievals and the AERONET inversions from almucantar scans for sixteen wavelengths in the 340- 2130 nm range and for seven wavelengths in the 340-1022 nm range focused on the coarse mode. The data correspond to $AOD_c>0.02$ and FMF<0.8 at 500 nm.

| | 16 wavelengths | | | 7 wavelengths | | | |
|---|---|---|---|---|---|---|---|
| Parameter | median difference | St.dev. | R | median difference | St.dev. | R | Number of measurements |
| $AOD_c$ | 0.008 | 0.014 | 0.98 | -0.002 | 0.010 | 0.98 | 30 |
| $C_{Vc}$ | -0.003 | 0.024 | 0.93 | 0.008 | 0.023 | 0.90 | 30 |
| $R_{eff}$ | 0.169 | 0.150 | 0.60 | 0.398 | 0.186 | 0.62 | 30 |
| $R_{Vc}$ | -0.275 | 0.390 | 0.64 | 0.253 | 0.484 | 0.16 | 30 |

### 3.3 Case studies for different aerosol types

In this section we focus on case studies of two aerosol types. First, we assess the performance of GRASP-AOD from PFR data under conditions where the predominant aerosol type is dust. We accomplish this by focusing on the Izaña site, where conditions are typically pristine except during dust episodes. Secondly, we focus on a highly unusual episode of long-range transport biomass burning smoke from the Canadian wildfires during the record-breaking year 2023 (Jain et al., 2024).

### 3.3.1 Dust

According to Barreto et al. (2022), values of AE<0.5 correspond to aerosol conditions dominated by dust or mixed cases. As dust is the main coarse-particle type in Izaña, restricting the dataset to AE<0.5, $AOD_c>0.05$ and fine mode fraction (FMF) $AOD_f/AOD<0.35$ should yield cases where dust is a significant part or even the predominant part of the total aerosol load. Therefore, we compared SD properties between GRASP-AOD from PFR data and AERONET inversions from almucantar

scans under these conditions. We summarize the results in Fig. 8 that includes the GRASP-AOD from PFR data and AERONET inversions from almucantar scans scatter plots for $R_{eff}$ and coarse mode parameters. $AOD_c$ shows excellent agreement and $C_{Vc}$ shows excellent correlation, though with some overestimation by from GRASP-AOD from PFR data (Fig. 8 a and b). $R_{Vc}$ showed no improvement compared to the general case (Fig. 8 c). $R_{eff}$ showed worse performance compared to the general case, as expected from the findings in Sect. 3.1.4. This can be explained by the fact that we found better performance for $R_{Vf}$ than $R_{Vc}$ and improvement under more restricted data. Therefore, under conditions where fine mode aerosol types are dominant, errors in $R_{Vc}$ affect $R_{eff}$ less, since $R_{eff}$ is more influenced by $R_{Vf}$. In conditions of mostly coarse particles (and thus larger $R_{eff}$), errors in $R_{Vc}$ affect $R_{eff}$ more significantly.

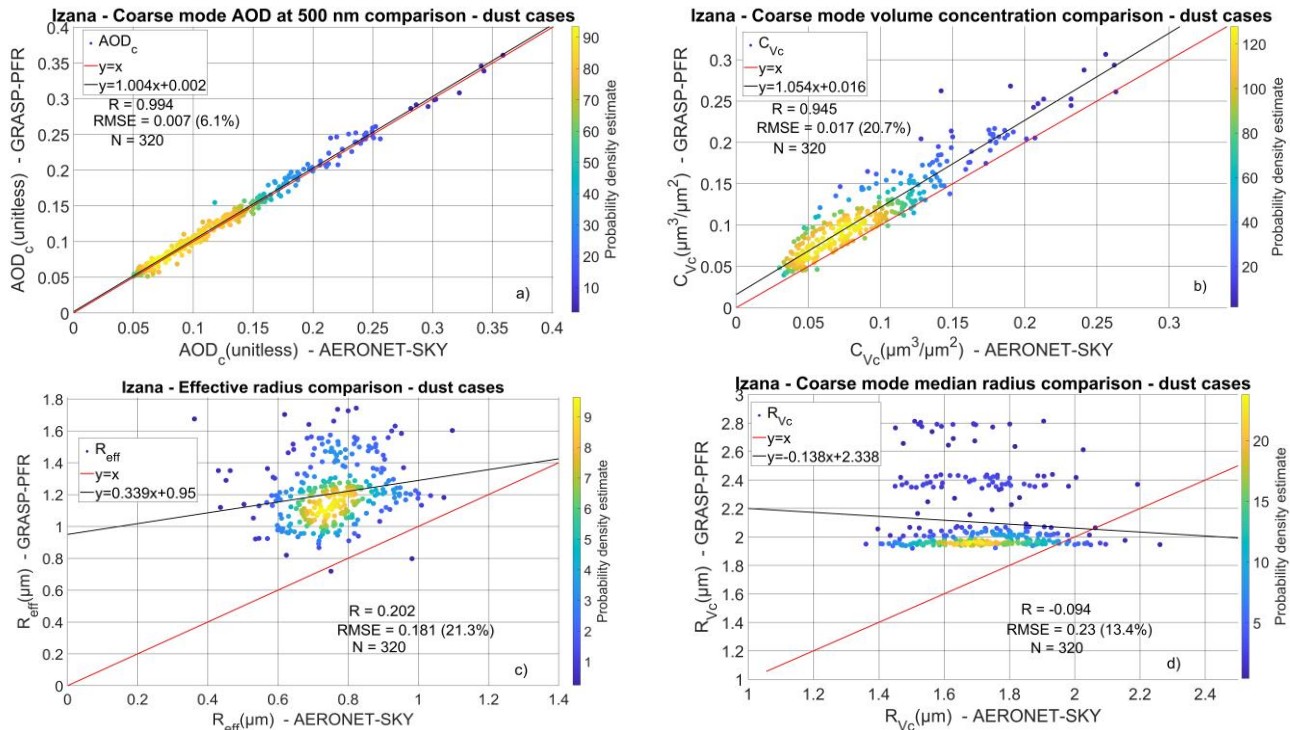

**Figure 8: Scatter plot of $AOD_c$ (a), $C_{Vc}$ (b), $R_{eff}$ (c) and $R_{Vc}$ (d) for the GRASP-AOD from PFR data and AERONET inversions from almucantar scans retrievals in Izaña corresponding to AE<0.5, $AOD_c$>0.05 and FMF<0.35 (which leads to large dust proportion in the overall aerosols).**

$R_{Vc}$ generally, shows very low sensitivity to AOD as expected (Torres and Fuertes, 2021). The GRASP-AOD from PFR data $R_{Vc}$ retrievals usually do not deviate significantly from the initial guesses, leading to very small parameter variation that is not present in the AERONET inversions from almucantar scans data (Fig. 8). Therefore, the multi-initial guess approach is not necessarily optimal for this parameter. Using prior information for the GRASP-AOD from PFR data retrievals -specifically a

520 single initial guess for $R_{Vc}$- may improve the results. This information (an $R_{Vc}$ average or climatology) can be derived by the best available source in each location and time period, such as ground based instruments that were measuring in that site for at least part of the period, satellite retrievals or modelling. Accordingly, we repeated the retrievals shown in Fig. 8 using the median $R_{Vc}$ from AERONET inversions from almucantar scans data (1.71 μm) as the initial guess. The results are presented in Fig. 9 that includes the same graphs as Fig. 8 using the single $R_{Vc}$ initial guess approach. The $AOD_c$ comparison shows no

significant differences. The $C_{Vc}$ from GRASP-AOD from PFR data is no longer systematically biased to larger values compared to AERONET inversions from almucantar scans. $R_{eff}$ remains biased to larger values, but the bias is reduced (intercept reduced from 0.95 to 0.74, RMSE from 21.3% to 13.4%) and $R_{Vc}$ is closer to AERONET inversions from almucantar scans as well (intercept reduced from 0.95 to 0.74 and RMSE from 13.4% to 3.6%).

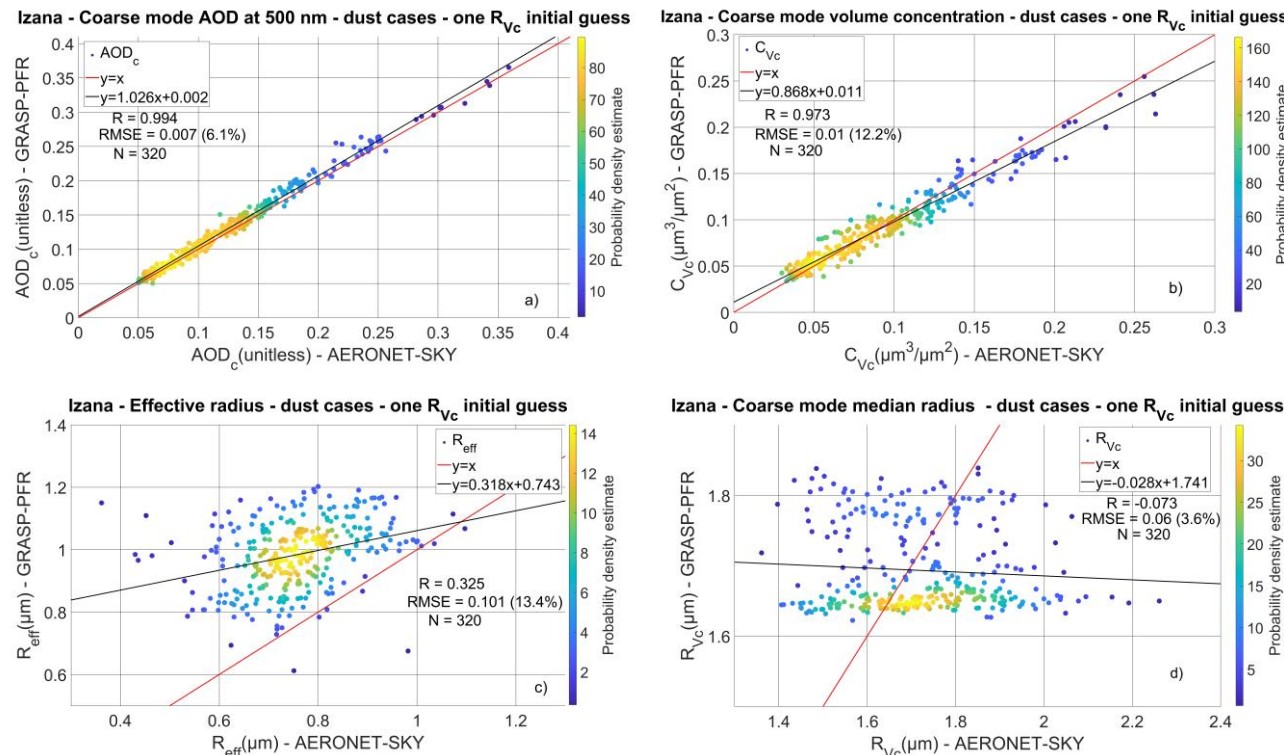

**Figure 9: Scatter plot of $AOD_c$ (a), $C_{Vc}$ (b), $R_{eff}$ (c) and $R_{Vc}$ (d) for the GRASP-AOD from PFR data and AERONET inversions from**
535 **almucantar scans retrievals in Izaña corresponding to AE<0.5, $AOD_c$>0.05 and FMF<0.35 and GRASP runs under one initial guess of $R_{Vc}$.**

### 3.3.2 Smoke episode from Canadian wildfires

During late September to early October 2023, long-range transport of smoke from the Canadian wildfires caused unusual AOD observations in several locations (Masoom et al., 2025) including Davos, where the highest AOD occurred at 500 nm rather than the shortest available wavelength. This led to negative AE in the UV and, in some cases, part of the visible spectrum. In Davos on 1 October 2023, this phenomenon was pronounced and observed by three different instruments (PFR, CIMEL, and BTS) (Fig. 10). In this section, we examine the SD characteristics associated with such aerosols and the extent to which GRASP retrievals using only AOD as input can reproduce those characteristics.

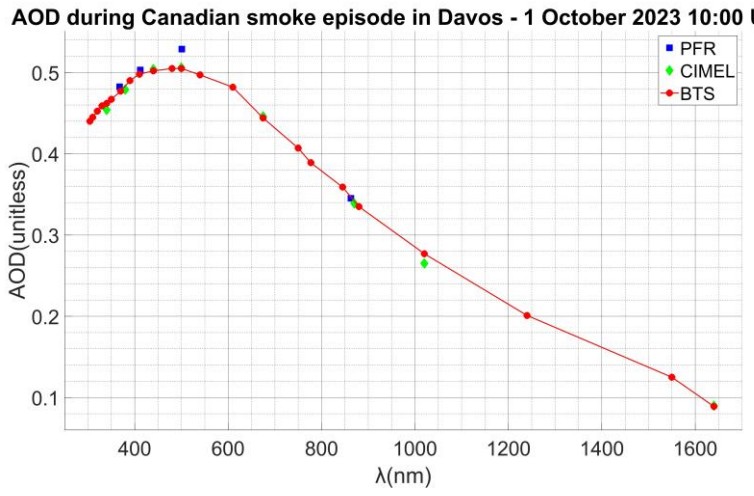

**Figure 10: The PFR, CIMEL and BTS AOD during a measurement of the unusual smoke episode in Davos during 1st of October 2023.**

During the smoke episode, we found two AERONET SD retrievals within 30 seconds of BTS and PFR AOD measurements. The SD shows that the vast majority of aerosols correspond to the fine mode, with a particularly high concentration and relatively narrow distribution. Moreover, the fine mode corresponds to unusually large radii (peak > 0.3 μm) as shown in the SDs of Fig. 11. Similar AOD behaviour was observed during the 2020 California wildfires, with similar SD characteristics (Eck et al., 2023).

To retrieve the SD from AOD, we did not use only the Torres and Fuertes, (2021) settings approach, since the low AE in that episode led to settings more appropriate for dust cases. The resulting output included strong overestimation of $C_{Vc}$ with $AOD_c$ > 0.2 at 500 nm and inversion residuals larger than our selection thresholds. Both AERONET inversions from almucantar scans and AERONET-SDA outputs showed $AOD_c$ < 0.003 during that smoke episode. Therefore, we also tested single retrievals with more general settings (Supplement Sect. S5) to check if the large coarse mode overestimation was indeed due to the settings. Although the alternative settings are not optimised for this particular episode, we reproduced the aforementioned SD characteristics for both PFR and BTS AOD. For BTS, we used a different wavelength selection than in Sect. 3.2 to include

additional UV channels due to the unusual AOD behaviour as the most interesting wavelengths in this episode are the shorter ones. However, both PFR- and BTS-based retrievals—regardless of AOD source—showed overestimation of $C_{Vf}$ and $R_{Vf}$ compared to AERONET (Fig. 11). PFR AOD led to higher $C_{Vf}$, due to the higher AOD at 500 nm (Fig. 10). CIMEL AOD yielded similar results to BTS for coincident measurements (Fig. S4). We observed similar episodes at several AERONET stations. One of them (Narsarsuaq in Greenland) included CIMEL AOD with a pronounced peak at wavelength above 340 nm (Fig. 12a) and SD observations within 1–8 minutes and low sky-radiance inversion residuals for AERONET SD (< 5 %; Holben et al., 2006). Retrieving SD with GRASP using CIMEL AOD and the same settings, we again reproduced the SD characteristics and overestimated $C_{Vf}$ and $R_{Vf}$ (Fig. 12b).

Finally, we tested the effect of wavelength selection on SD retrievals using only BTS AOD. We applied four wavelength selections (Sect. S5): two spanning UV to near-IR, one excluding UV wavelengths, and one excluding IR wavelengths. All selections yielded similar SDs. Excluding UV resulted in a larger $C_{Vf}$ difference compared to the others, but the difference remained small (Fig. 11).

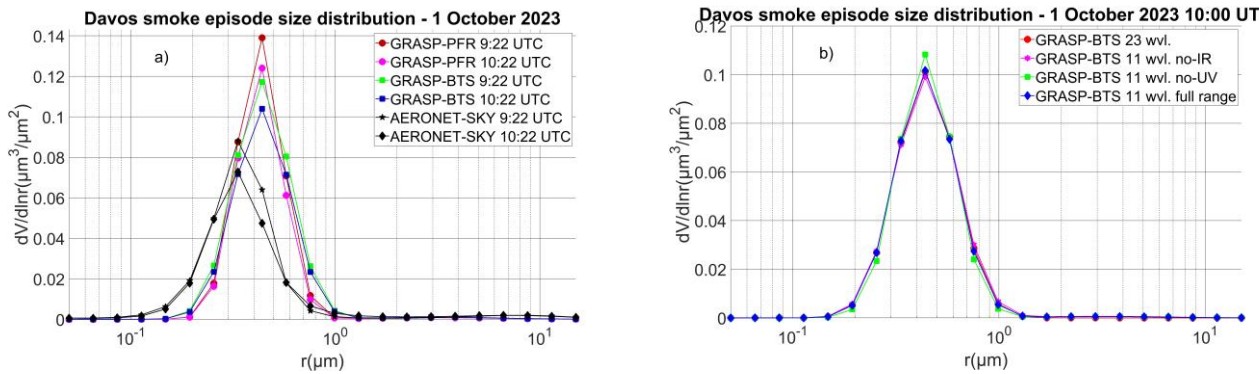

**Figure 11: The SDs of AERONET (black lines) and GRASP retrievals (coloured lines) from PFR and BTS for two different common measurements (a) and the BTS retrievals for different wavelength selections (b).**

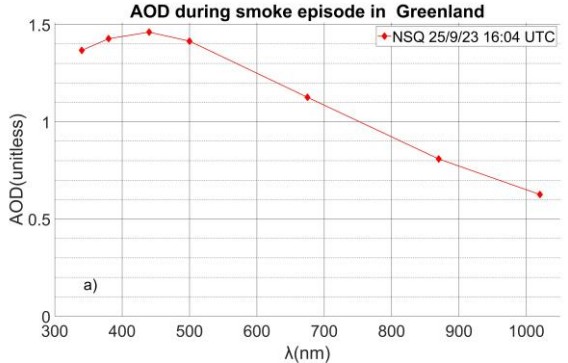

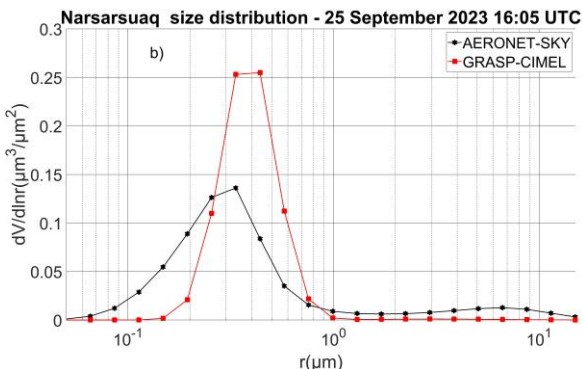

Figure 12: (a) CIMEL AOD per wavelength at Narsarsuaq during the record-breaking Canadian wildfires. (b) GRASP-CIMEL size distribution retrievals (red lines) and the closest AERONET size distributions (black lines) at Narsarsuaq (b) during the record-breaking Canadian wildfires.

## 4 Discussion

In the previous sections, we explored the capabilities of retrieving bimodal aerosol SD parameters using only AOD observations and the inversion model GRASP. In the first part of the study (Sects. 3.1.1, 3.1.2 and 3.1.3), we validated the GRASP-AOD from PFR data retrievals against AERONET products (AERONET inversions from almucantar scans and AERONET-SDA). Our main findings are consistent with the validation study of Torres and Fuertes, (2021), which used AERONET AOD to retrieve SD parameters with GRASP. Of course, differences arise owing to the different locations and datasets. Torres and Fuertes, (2021) employed many sites, including several with higher aerosol loads, yielding thousands of data points with AOD>0.4 at 440 nm, where inversion uncertainties are lower. In contrast, the GAW-PFR network includes fewer stations, mostly under pristine conditions. Therefore, in the present study it is of particular interest the performance of GRASP under low aerosol loads. Moreover, the availability of parallel observations from both AERONET and GAW-PFR, and the PFR's one-minute temporal resolution, allowed a different validation methodology. Another difference is instrument spectral range: CIMEL covers seven or eight wavelengths (340-1020 nm or 340-1640 nm), whereas PFR covers four (368-862 nm). A key question was whether the PFR's information content suffices to retrieve SD parameters with similar quality. Our findings show that the PFR's spectral range and resolution are sufficient to retrieve aerosol SD parameters using GRASP—except $R_{Vc}$, as also noted by Torres and Fuertes, (2021).

The best GRASP-AOD from PFR data performance is evident in $AOD_f$ and $AOD_c$ retrievals, where we observe the highest correlations and the lowest (or among the lowest) relative differences against AERONET inversions from almucantar scans. Most differences between GRASP-AOD from PFR data and AERONET-SDA (78.1% for $AOD_f$ and 82.4% for $AOD_c$) lie within AERONET-SDA uncertainty estimates. AOD modal separation performed excellently for AOD as low as 0.02 at 440 nm across different aerosol types (dust, Sect. 3.3.1; long-range smoke, Sect. 3.3.2).

The volume concentrations ($C_{VT}$, $C_{Vf}$ and $C_{Vc}$) showed good correlations between GRASP-AOD from PFR data and AERONET inversions from almucantar scans (R>0.85) with relative median differences of 6.6-39.9 %. $C_{Vf}$ is very low for most of measurements in these sites, so the relative difference is larger. Relative standard deviations and RMSEs of the linear fit between GRASP-AOD from PFR data and AERONET inversions from almucantar scans, however, ranged 47.6-62.5 %. $C_{Vc}$ from GRASP-AOD from PFR data was overestimated for predominantly dusty data; this bias disappeared via a carefully selected initial guess of $R_{Vc}$.

Volume radii ($R_{eff}$, $R_{Vf}$, $R_{Vc}$) showed variable performance in the comparisons between GRASP-AOD from PFR data and SKY-AER, depending on the parameter and conditions. $R_{Vf}$ is a parameter with low variability, which results in a low correlation factor even for small relative differences. It shows the lowest relative standard deviation (27.9%) if we exclude $AOD_f$. However, only 15.8% of the differences between GRASP-AOD from PFR data and AERONET inversions from

almucantar scans were within the AERONET inversions from almucantar scans uncertainties. Our findings showed that the largest deviations occur at low AE. Restricting the analysis to AE>1, AOD>0.1 and $AOD_c$>0.04 at 500 nm, we showed that there is significant increase in the correlation factor and decrease in the variance of the differences between GRASP-AOD from PFR data and AERONET inversions from almucantar scans. $R_{Vc}$ on the other hand does not improve when restricting the data to specific conditions due to its low sensitivity in PFR AOD. For $R_{Vc}$ only 9.5% of the differences lie within the AERONET inversions from almucantar scans uncertainties. However, it can be improved by providing to GRASP a single $R_{Vc}$ initial guess close to the average of AERONET inversions from almucantar scans. $R_{eff}$ showed a variable performance similar to $R_{Vf}$ (significant improvement for AE>1, AOD>0.1 and $AOD_c$>0.04). $R_{eff}$ is affected by the performance of $R_{Vf}$ and $R_{Vc}$ depending on the concentrations. When the aerosol load is mostly fine particles, $R_{eff}$ will be affected mostly by the accuracy of $R_{Vf}$. When the aerosol load consists of mostly coarse particles, is affected more by the accuracy of $R_{Vc}$. As a result, when AE is large enough the more accurate $R_{Vf}$ and low weight of $R_{Vc}$ result in a better $R_{eff}$ estimation. In the opposite case the accuracy is largely reduced. Conversely, accuracy falls when AE is low, unless the $R_{Vc}$ initial guess approximates reality.

Regarding the effect of refractive index assumption, we found that using a single value of complex refractive index as input to GRASP showed no significant differences compared to the use of climatologies depending on the month and the station. This was a result of the fixed value being close to the average of the climatologies and showing small enough deviations from them not to largely affect the statistics of the retrieval comparisons. This is a useful finding for the application of GRASP to GAW-PFR network since most of the sites do not include a co-located AERONET instrument to provide consistently long-term local observations for the refractive index. In such stations the refractive index selection and the $R_{Vc}$ initial guess could be a fixed value or climatologies derived from satellite data (Chen et al., 2020), modelling (Taylor et al., 2014), or in-situ ground-based or airborne observations (Espinosa et al., 2017; Espinosa et al., 2019).

To assess potential improvements in the retrieval of $R_{Vc}$ with larger spectral range and generally the effect of wavelength selection on the retrievals of the SD parameters, we used AOD datasets corresponding to two different wavelength selections and two AOD calculation methods. One wavelength selection included seven wavelengths similar to CIMEL in the 340-1022 nm range and the second sixteen wavelengths in the 340-2130 nm range. Our results showed that the increased spectral range had minor effect for most parameters, but $R_{Vc}$ retrievals are no longer mostly stuck very close to the initial guess (as it happens for the PFR and CIMEL standard version spectral ranges). The additional infrared channels create some sensitivity of $R_{Vc}$ to the AOD values.

Using AERONET inversions from almucantar scans retrievals as reference, we found either positive or negative effects when using the larger spectral range in the aerosol characterization. All parameters except $R_{Vc}$ showed small differences in terms of GRASP-AOD from BTS data and AERONET inversions from almucantar scans correlation between the different wavelength selections ($\Delta R$<0.15 except when comparing. The main positive effects of using a larger spectral range were the substantial increase of the correlation with AERONET for $R_{Vc}$ and a reduced standard deviation of the radii differences between GRASP-AOD from BTS data and AERONET inversions from almucantar scans compared to the selection of fewer wavelengths. However, this was accompanied by an increased median difference. Possible explanations for this increased bias can be related

to the uncertainties of AOD and the GRASP retrievals from AOD or how GRASP responds for the wavelength selection, but in this particular case, for $R_{Vc}$, the following explanation is probably more important. Using an $R_{Vc}$ initial guess close to the AERONET inversions from almucantar scans average and a smaller spectral range results in GRASP-AOD from BTS data

$R_{Vc}$ retrievals close to that $R_{Vc}$ average. If the variability of $R_{Vc}$ in a particular place or dataset is low and we provide that average to GRASP as an initial guess, then using AOD with the smaller spectral range (seven wavelengths) will result in GRASP remaining close to reality. If instead we use the larger spectral range to retrieve $R_{Vc}$, some retrievals will deviate more from the initial guess despite the proximity of the initial guess to reality, as the retrieval retains some uncertainty and the model needs to fit six SD parameters during the inversion. This may result in more accurate $R_{Vc}$ retrieval when using the smaller

spectral range selection compared to the larger. Smaller accuracy in $R_{Vc}$ retrieval will result in lower accuracy of $R_{eff}$ retrieval, to some extent may affect $C_{VT}$, $AOD_c$ and $C_{Vc}$ as well. However, we used a small number of AERONET measurements that correspond to a station of low aerosol load that corresponds mostly to the fine mode. We also used an initial guess based on the presence of AERONET instrument. This results in more accurate GRASP retrievals when using the selection of seven wavelengths than expected for cases where less information is available or the coarse mode aerosols show larger load and

variability. This increased accuracy can 'hide' potential benefits of the increased spectral range and our conclusions cannot be generalized. The median $R_{Vc}$ of AERONET inversions from almucantar scans in the data used for comparison of Table 9 is 2.25 µm. For GRASP-AOD from BTS data using AOD at seven wavelengths, the median $R_{Vc}$ and the used initial guess is 1.75 µm. The median GRASP-AOD from BTS data retrieved $R_{Vc}$ using AOD at sixteen wavelengths, including 2130 nm, is 1.67 µm. Therefore, we can explain the decreased bias of $R_{Vc}$ compared to AERONET when using seven wavelengths through the

proximity between the AERONET average of this dataset and the initial guess. The median $R_{Vc}$ of the dataset restricted to $AOD_c>0.02$ and FMF<0.8 at 500 nm is 1.56 nm for AERONET, 1.81 nm for BTS with seven wavelengths selection and 1.37 nm for BTS with sixteen wavelengths. GRASP-AOD from BTS data AERONET inversions from almucantar scans In Fig. 8 (panel c) we can see that $R_{Vc}$ when using sixteen wavelengths showed a significant portion of low values (<1.3 µm), but the majority of the data corresponds to values closer to the more usual range of AERONET inversions from almucantar scans

values. We found no clear correlation between $R_{Vc}$ from GRASP-AOD from BTS data at sixteen wavelengths and AOD, AE or $AOD_c$ except that $R_{Vc}<1.4$ µm rarely appeared for $AOD_c>0.23$ or AOD>0.26 at 500 nm. Therefore, we cannot attribute the low values and low accuracy of $R_{Vc}$ from GRASP-AOD from BTS data using the higher spectral range to the increased $AOD_c$ of the retained measurements compared to the other 33 measurements. The reason for this discrepancy is unclear, but the sample size is particularly small. Yet, the increase in correlation factors of $R_{Vc}$ between AERONET and BTS is particularly

encouraging (full dataset: R=-0.04 for seven wavelengths of AOD and R=0.73 for sixteen. Increased coarse mode dataset: R=0.16 for seven wavelengths of AOD and R=0.64 for sixteen). Therefore, it is not clear how the conditions affect the GRASP-AOD from BTS data accuracy due to the limited data with sufficient coarse mode particles.

The comparisons between GRASP-AOD from BTS data and AERONET inversions from almucantar scans comparison showed good consistency in most cases (Tables 6, 9 and 10) with the comparisons between GRASP-AOD from PFR data and

AERONET inversions from almucantar scans comparison, although the first corresponds to only a small number of

measurements (up to 63) that were not included in the GRASP-AOD from PFR data and concern only one of the four selected stations. PFR and BTS also show differences between their AOD, which were limited by filtering the BTS AOD using the PFR as reference (Sect. 2.4). Retrievals from AOD at sixteen wavelengths showed reduced standard deviation of the $R_{Vc}$ differences with AERONET inversions from almucantar scans and higher R compared to both seven-wavelength BTS and

PFR data, which further shows that the additional spectral range may improve the retrieval of $R_{Vc}$.

In the case of the unusual smoke episode we studied, we found that GRASP can be particularly sensitive to the settings in such conditions. Using single retrievals instead of the multi-initial guess approach and more general settings, it was enough to reproduce with GRASP the characteristics of this rare aerosol SD using AOD from three different instruments, but with an overestimation in the concentration and radius of the aerosols. Testing this in one more AERONET station, we found consistent

results. Using the spectral range and resolution of BTS, we were able to test the effect of wavelength selection in this case. The SD did not show sensitivity to the wavelength selections during this smoke episode in Davos.

## 5 Summary and conclusions

In this study, we used the GRASP model to retrieve aerosol SD properties and AOD modal separation from AOD observations at four locations, which include instruments from both the GAW-PFR and AERONET networks. We used as reference the

AERONET output parameters (AERONET inversions from almucantar scans and AERONET-SDA).

The AOD comparisons between PFR and AERONET AOD (either AERONET data from direct sun and AERONET inversions from almucantar scans) in the four sites showed good agreement, with all median differences and standard deviations being < 0.01 (the AOD uncertainty at air mass 1).

Separation of AOD into AOD of fine and coarse mode of GRASP-AOD from PFR data was the output parameter that showed

the best performance. It showed excellent correlation with both AERONET inversions from almucantar scans and AERONET-SDA (R>0.98) (Table 6) the median and standard deviation of the differences were within the uncertainties of the AERONET-SDA retrievals and more than 78% of the points within the uncertainties as well.

Volume concentration retrievals showed very good correlation between GRASP-AOD from PFR data and AERONET inversions from almucantar scans (R~0.88-0.96). The relative median differences were above 20% for $C_{VT}$ and $C_{Vc}$, while for

$C_{Vf}$ the median difference was 6.6%. This can be explained by the stronger presence of fine mode aerosols in the selected sites. The relative standard deviations were above 45% showing larger relative variance in the comparisons compared to the other parameters except $R_{eff}$. This is related both to the retrieval uncertainties and the low aerosol loads in those stations.

The radii retrievals showed lower correlation (R~0.21-0.82) compared to the AOD separation and the concentrations. $R_{eff}$ showed overestimation and large variance at larger $R_{eff}$ values (>0.5-0.7 μm) or smaller AE (AE<1-1.3). $R_{Vf}$ and $R_{Vc}$ showed

the lowest correlation and less than 16% of the differences between GRASP-AOD from PFR data and AERONET inversions from almucantar scans were within the AERONET inversions from almucantar scans uncertainties. $R_{Vf}$ has low variability and the correlation factor is not so representative indicator of its performance. It can be significantly improved by limiting the

datasets to observations corresponding to AOD at 500 nm > 0.1 and AE > 1. For $R_{Vc}$ we found no improvement by limiting the datasets to certain conditions. $R_{Vc}$ showed improvement when we provided an $R_{Vc}$ initial guess closer to the reference median $R_{Vc}$. For the radii, no more than 15% of the differences were within the AERONET inversion uncertainties.

Comparing our results with the equivalent ones from Torres and Fuertes, (2021) for GRASP retrieval using AERONET AOD, we found good consistency between the two studies, despite the differences in the instrument characteristics, site selection and intercomparison methodology. The results were also not significantly affected by the retrieval under our selections of complex refractive indices, which shows that the aerosol size characterization is possible without the presence of an instrument dedicated to the refractive index observation if the selection is not too far from reality.

Using different wavelength selections from BTS AOD, we found that GRASP-AOD from BTS data retrievals are affected by the AOD wavelength selection. We also found that $R_{Vc}$ shows some sensitivity to AOD when we include long enough wavelengths and the correlation factor increases in that case. However, using an $R_{Vc}$ initial guess close to the average of reality under low variability of $R_{Vc}$ may reduce the $R_{Vc}$ bias compared to the use the larger spectral range. Regarding the other parameters, we found no consistent improvement. The performance was similar to the performance of GRASP-AOD from PFR data retrievals. Therefore, the larger spectral range increases the algorithm's capabilities for $R_{Vc}$ retrieval, which is the main limitation of the smaller spectral range and the same time seems to preserve the performance in the other parameters. Our results correspond to limited AERONET data only on one site with mostly pristine conditions and fine mode aerosols, which limits our capabilities to derive more concrete conclusions. Additional sites and more research are required to achieve more solid conclusions regarding the benefit of the larger spectral range in such retrievals. Assessing the differences between the GRASP-AOD from PFR data and GRASP-AOD from BTS data comparisons with AERONET inversions from almucantar scans, further supported the aforementioned conclusions.

Focusing on conditions where the predominant aerosol type is dust, we found consistent results with the findings found above. As dust particles are mainly large enough to correspond in the coarse mode, such cases lead to small AE and large $R_{eff}$, which results in less accurate retrieval of the radii. The results were good as expected for AOD modal separation and volume concentrations. Again, $R_{eff}$ and $R_{Vc}$ showed improvement through the selection of $R_{Vc}$ initial guess based on $R_{Vc}$ AERONET inversions from almucantar scans retrievals.

Finally, in the case of unusual AOD observations during an episode of smoke from the Canadian wildfires in 2023 (negative AE up to 500 nm), we found that the aerosol SD is also unusual. SD included mostly fine mode particles that were unusually large for their type and present in high concentration. GRASP retrievals using AOD from different instruments (PFR, CIMEL and BTS) successfully reproduced these characteristics, but showed overestimation of concentration and radius.

*Code availability*. The GRASP software and documentation is available at the relevant GRASP-SAS website: https://www.grasp-open.com/


*Data availability*. The PFR AOD data and GRASP-AOD from PFR data retrievals are available in Zenodo (Karanikolas et al., 2024). The BTS AOD is available in communication with authors.

The CIMEL AOD data are available from https://aeronet.gsfc.nasa.gov/. The following references correspond to each site:

Davos: Kouremeti et al., 2024.

Hohenpeissenberg: Mattis et al., 2024.

Izaña: Goloub et al., 2024.

Lindenberg: Becker et al., 2024.

*Author contribution*. AK analysed the data and wrote the paper with contributions from the co-authors. AK and SK

conceptualized the study. BT contributed to the algorithm development and modification for GRASP retrievals from AERONET and GAW-PFR AOD. BT, MM and MHG contributed to the understanding and operation of the GRASP model. MM contributed to the GRASP configuration for retrievals from the BTS spectroradiometer AOD. NK and SK contributed to the PFR sun photometer data provision. JG contributed to the BTS data provision. LD contributed to the PFR sun photometer data provision in Lindenberg. All authors were involved in the interpretation of the results and reviewing the paper.

*Competing interests*. The authors declare that they have no conflict of interest.

*Financial support*. This research has been supported by the European Metrology Programme for Innovation and Research (grant no. 19ENV04 MAPP) and COST (European Cooperation in Science and Technology) under the HARMONIA (International network for harmonization of atmospheric aerosol retrievals from ground-based photometers), action no. CA21119.


*Acknowledgments*. The authors would like to acknowledge the ESA project QA4EO, grant no. QA4EO/SER/SUB/09 and HARMONIA (International network for harmonization of atmospheric aerosol retrievals from ground-based photometers), action CA21119.

Angelos Karanikolas has been supported by the European Metrology Program for Innovation and Research (EMPIR) within

the joint research project EMPIR 19ENV04 MAPP "Metrology for aerosol optical properties". EMPIR is jointly funded by the EMPIR participating countries within EURAMET and the European Union.

Stelios Kazadzis, Angelos Karanikolas and Natalia Kouremeti would like to acknowledge the ACTRIS Switzerland project (Aerosol, Clouds and Trace Gases Research Infrastructure – Swiss contribution) funded by the Swiss State Secretariat for Education, Research and Innovation.

Angelos Karanikolas would like to acknowledge Dr. Juan Carlos Antuña Sánchez for the IT support regarding access and operational issues related to the GRASP software.

The authors would like to acknowledge Mr. Virgilio Carreño-Corbella, Mr. Ramón Ramos, the observers and maintenance team, for continuously supporting the operation and upkeep of the Izaña observations and Dr. Frank Wagner for the observations in Hohenpeissenberg. Also, Dr. Africa Barreto, Dr. Ralf Becker and all PIs and local operators in the four stations over the years.

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

Appendix

Table A1: List of abbreviations.

| | |
|---|---|
| GAW-PFR | Global Atmospheric Watch-Precision Filter Radiometer |
| AERONET | Aerosol Robotic Network |
| WMO | World Meteorological Organization |
| GRASP | Generalized Retrieval of Atmosphere and Surface Properties |
| PFR | Precision Filter Radiometer |
| CIMEL | CIMEL CE318-TS sun and sky photometer |
| BTS | The array spectroradiometer 'BiTec Sensor'. |
| POM | PREDE-POM sun and sky radiometer |
| FRC | Filter Radiometer Comparison |
| AOD | Aerosol Optical Depth |
| AE | Ångström Exponent |
| DSI | Direct Solar Irradiance |
| $AOD_f$ | Fine Mode Aerosol Optical Depth |
| $AOD_c$ | Coarse Mode Aerosol Optical Depth |

| | |
|---|---|
| $C_{VT}$ | Total Volume Concentration |
| $C_{Vf}$ | Fine Mode Volume Concentration |
| $C_{Vc}$ | Coarse Mode Volume Concentration |
| $R_{eff}$ | Effective Radius |
| $R_{Vf}$ | Fine Mode Volume Median Radius |
| $R_{Vc}$ | Coarse Mode Volume Median Radius |
| $\sigma_{Vf}$ | Fine Mode Geometric Standard Deviation |
| $\sigma_{Vc}$ | Coarse Mode Geometric Standard Deviation |
| FMF | Fine mode fraction of AOD |
| RRI | Real part of the aerosol Refractive Index |
| IRI | Imaginary part of the aerosol Refractive Index |
| SSA | Single Scattering Albedo |
| SD | Aerosol Size Distribution |
| GRASP-AOD | Aerosol Properties Retrieval(s) using the AOD as input. |
| AERONET-SDA | AERONET retrievals of aerosol optical depth modal separation using the spectral deconvolution algorithm. |

| | |
|---|---|
| SZA | Solar zenith angle |
| FoV | Field-of-View Angle |
| FWHM | Full-Width-at-Half-Maximum |
| St.dev. | Standard Deviation |
| P95th | 95th percentile |
| P5th | 5th percentile |
| R | Pearson correlation factor |
| $R^2$ | Coefficient of determination |
| RMSE | Root mean square error |
| AOD-obs | AOD retrieved directly from the direct spectral irradiance measured by an instrument. |
| AOD-ext | AOD estimated by the Ångström law after calculation of the Ångström exponent and turbidity coefficient using observed spectral AOD. |
| UV | Ultraviolet |
| IR | Infrared |
| $O_3$ | Ozone |
| $H_2O$ | Water vapour |

| | | |
|---|---|---|
| $CH_4$ | Methane | |
| $N_2O$ | Nitrous oxide | |

1100