# Peer review of "Retrieval of Aerosol Properties from Aerosol Optical Depth Measurements with High Temporal Resolution and Spectral Range"

_EGUsphere, 2025_

## Author Comment (AC1)

**Author's response to referee #1**

The authors would like to thank the referee for providing helpful and constructive comments. You can find our response below each comment.

1. Please add the specific latitude and longitude information of the four stations in Section 2.1.

Added the requested in information in the location descriptions.

2. In ground-based instrument observations, calibration is an important factor affecting inversion accuracy. In Section 2.1, many instruments (CIMEL, PFR, BTS) in the manuscript have conducted continuous observations for many years. Please provide additional information on how often these instruments are calibrated.

Added information about the calibration procedures in the sections 2.1.1, 2.1.2 and 2.1.3.

The PFRs used in Davos are part of the AOD world reference triad. The stability of the GAWPFR triad is monitored by co-located instruments calibrated with the Langley plot method at Izana, Tenerife and Mauna-Loa, Hawaii. The average frequency of stability checks is 6 months. The instruments from these two sites are calibrated again with the Langley plot method the first month after their return to the Izana or Mauna Loa. The stability check in Davos continues for a period of at least 15 days with clear Sun. Re-calibration of the reference instruments is implemented when necessary, according to the results of the stability checks.

The PFR operated at Izana is calibrated with the Langley plot method with an average recalibration rate of 6 months depending on the stability of the instrument and operation conditions (mainly alignment to sun).

The PFR at Hohenpeissenberg (GAW-PFR site) is exchanged every 2-3 years and the data is reprocessed based on the two calibrations initial and upon return to PMOD/WRC.

The Lindenberg (associated station) PFR is operated by Deutscher Wetterdienst (DWD and the instruments are sent to PMOD/WRC for calibration every 2 years, following a rotation schema among the DWD sites.

AERONET CIMELs from Izana are calibrated on site every 6 months with calibration transfer against another CIMEL acting as reference. The two reference instruments are calibrated with the Langley plot method one at a time in rotation every three months.

The calibration of the Davos CIMEL is performed at the Laboratoire d'Optique Atmosphérique - Université de Lille (LOA) by comparison with master instrument. Since 2018 a calibrated instrument is provided for exchanging the field instrument on a yearly basis. Previously the instrument was shipped to LOA for calibration after 1-1.5 years of operation.

Lindenberg and Hohenpeissenberg include two CIMELs that are transported to University of Valladolid alternatingly one at a time, approximately once per year to be calibrated with calibration transfer from a master instrument.

BTS are calibrated yearly though, measurements of irradiance from 1000W lamps in PMOD/WRC (Davos) optic lab. Every 2 months measurements of irradiance from portable 200 W lamps are used to monitor the stability of the BTS.

3. In Section 2.1.3, it is mentioned that BTS has two detector arrays, and there is a part of the repeated spectra (300–1050 nm and 950–2150 nm). How are the repeated spectral ranges, such as 1020 nm, selected and processed?

The solar spectra are combined at 1000 nm. All channels at shorter wavelengths are used from BTS measuring the shorter wavelengths (BTSVIS), while all channels with wavelengths longer than 1000 nm are used from BTS measuring only in the infrared (BTSIR). An overlapping region between 980 and 1020 nm is used as diagnosis but not corrected. However, the ratio between BTSIR to BTSVIS is usually well within 1%.

Added in the manuscript the information that all selected wavelengths above 1000 nm are taken from BTSIR.

4. In Figure 1, there are slight differences in the criteria used for the observed and extrapolated AODs for 7 and 16 bands. For example, in the 7 bands, AOD-obs uses 0.015, but AOD-ext uses 0.011, and so on. Why not use the same filter criteria for the same number of channels?

As it happened when we reduced the number of wavelengths, using the extrapolated AOD resulted in easier convergence for the model, which reduced the inversion residuals compared to the use of observed AOD. These smaller residuals do not necessarily result in more accurate aerosol properties (which we can observe in the comparisons with AERONET as well). Therefore, we used stricter selection criteria for the extrapolated AOD as we did for the reduced number of wavelengths (PFR compared to CIMEL and 7 BTS wavelengths compared to 16).

However, this is no longer relevant as we removed the GRASP retrievals from the AOD-ext as response to comments from another referee.

5. What does "P95-P5" mean in Table 1? Please explain.

This is the difference between the 95th and 5th percentiles. Added explanation in the caption. P95 and P5 will now appear on the abbreviations table in the appendix.

6. In Figures 6 (c) and (d), what does "a" mean in the last item of the title filtering conditions? Is it AE? Please check.

Yes, corrected.

7. For better comparison, it is recommended to unify the horizontal axis range of the upper and lower groups of Figure 8.

Unified the range of horizontal axes.

8. In the table, some table parameter columns (Table 2) use bold fonts, and some table parameter columns (Tables 3-6) use normal fonts. Please check and unify them according to the requirements of the journal.

**Corrected.**

9. In Section 4 (Line 635), the manuscript mentions that there is an underestimation of radius and concentration in the GRASP inversion results, but according to Figure 12 and the previous description, it seems to be an overestimation. Please check.

Corrected to 'overestimation'.

**Modifications not requested by the referee:**

1) Added a citation regarding the Canadian wildfires in line 482 (Section 3.3.2).

Masoom, A., Kazadzis, S., Modini, R. L., Gysel-Beer, M., Gröbner, J., Coen, M. C., Navas-Guzman, F., Kouremeti, N., Brem, B. T., Nowak, N. K., Martucci, G., Hervo, M., and Erb, S.: Long range transport of Canadian Wildfire smoke to Europe in Fall 2023: aerosol properties and spectral features of smoke particles, EGUsphere [preprint], https://doi.org/10.5194/egusphere-2025-2755, 2025.

**Rephrased in lines 482-483:**

'Canadian wildfires caused unusual AOD observations (Masoom et al., 2025), where the highest AOD occurred at 500 nm rather than the shortest available wavelength, leading to'

**To:**

- 'Canadian wildfires caused unusual AOD observations in several locations (Masoom et al., 2025) including Davos, where the highest AOD occurred at 500 nm rather than the shortest available wavelength. This led to'
- 2) Line 608: changes 'retrained' to 'retained'.
- 3) Added acknowledgements about local operators for the PFR and CIMEL datasets.

**Major modifications as response to another referee:**

- 1) Added information in Sections 3.1 and S1 about the comparisons of GRASP-AOD retrievals from PFR AOD and AERONET for the full datasets (without the selection criteria for AOD differences, AODf/c and AE).
- 2) Removed from Sections 2.4, 3.2 and related discussion all mentions and results of GRASP-AOD retrievals from interpolated or extrapolated AOD using the Ångström approximation. We retained Fig. 7.
- 3) Shortened the introduction and discussion about BTS spectroradiometer related retrievals to be clearer and more focused on the main objectives of the manuscript.

---

## Author Comment (AC2)

**Author's response to referee #2**

The authors would like to thank the referee for providing helpful and constructive comments. You can find our response below each comment.

**Major comments:**

1. There is an overall strategy to "improve" the comparisons by removing "bad" data and fine-tuning the retrievals. Data in which the AOD differences are "too high" are removed. (line 271). Interpolated AOD from BTS is removed if R2<0.8. (line 268). The authors retain the best cases only, ignoring all the difficulties that yield to discrepancies and that such circumstance will always come together with real data. Line 301: how could differences be large if all discrepant data points were previously removed?

The same applies to the retrievals: thresholds to residuals are changed ad-hoc to "improve" results (Figure 1). And the initial guess (line 466, 541, 550, 594), the settings (line 495), the wavelength range (line 499) or the refractive index are changed to adapt to certain cases. Or the dataset is restricted to the most favorable conditions (line 366, 546).

And my quotes in "improve" are put to emphasize that this is not a real improvement, just artificial clean-up of presumed outliers..

We find this concern understandable and acknowledge that the manuscript lacked sufficient justification of the methodology in certain cases. We list below explanations and modifications to the manuscript for each one of these matters. The various selection criteria are included in different parts of the manuscript, for different reasons, so we consider it necessary to treat them in a case-by-case manner. Therefore, the response below is divided into 5 parts referring to each one of the issues raised by the referee (such as aerosol optical depth (AOD) differences, inversion residuals, settings). Each part includes separately the idea behind the selection criteria and any modifications we implemented. In certain cases, we also included additional information, figures or tables to better clarify the situation. The responses of 'minor' comments related to this major comment are also merged into the response of the present comment.

**1) AOD differences**

**a) CIMEL and PFR**

**Explanation**

GRASP inversions output uncertainties are linked with two main sources:

- the inversion methodology
- AOD input uncertainties

By comparing the GRASP output with the AERONET one, it is not easy to separate the two sources, so cases of large differences in the size distribution parameters may stem either from the inversion methodology uncertainties or large errors in AOD. AOD differences can be the

result of various reasons. Some are related to the limitations of retrieving the AOD such as: Calibration uncertainties, AOD algorithm post processing differences, measurement aspects (e.g. the effects of the signal random noise or the field of view of the instruments). Other factors resulting in AOD differences are the limitations of cloud screening algorithms, insufficient instrument cleaning, imperfect synchronisation of measurements under highly variable conditions or instrument technical issues. An analysis to explain and quantify AOD differences among PFR and CIMEL instruments and the effect of each factor has been presented in Cuevas et al., 2019.

Our aim was to better assess the performance of the inversion methodology for the characteristics of the instruments used, under the typical uncertainties of the observations by minimising the effect of factors such as cloud contamination and imperfect synchronisation, so we selected the data within certain thresholds of AOD differences.

In order to provide a better insight, we repeated the comparisons without the selection criteria and decided to include results and information both from filtered and unfiltered data (shown in Table R1).

Table R1: Difference of statistics of the comparisons between GRASP-AOD from PFR data retrievals and AERONET radiance inversions for different data sections (all data minus filtered according to the pre-print). We also include the AOD at 500 nm comparison between the PFR and AERONET radiance inversions, the correlation factor (R) and the relative median difference compared to the median of each parameter from the reference dataset (PFR for AOD, AERONET for every other parameter). The total number of measurements ('unfiltered', AOD>0.02 at 440 nm) is 5400.

| Parameter               | ΔMedian difference | ΔSt.dev. | ΔR     | ARelative median difference (%) | Amedian of the parameter |
|-------------------------|---------------------------|----------|--------|---------------------------------|--------------------------|
| AOD 500 nm              | 0.000                     | 0.004    | 0.002  | -0.9                            | -0.018                   |
| $AOD_f500\;\text{nm}$   | 0.001                     | 0.002    | 0.004  | 1.3                             | -0.015                   |
| AOD c 500 nm | 0.000                     | 0.000    | -0.002 | -21.6                           | -0.033                   |
| $C_{VT}$                | 0.001                     | 0.000    | 0.013  | -0.8                            | -0.007                   |
| $C_{ m Vf}$             | 0.001                     | 0.000    | 0.013  | 4.4                             | -0.002                   |
| $C_{Vc}$                | 0.006                     | -0.004   | 0.006  | -13.7                           | -0.026                   |
| $R_{eff}$               | -0.012                    | -0.021   | -0.031 | -4.7                            | -0.036                   |
| $R_{\mathrm{Vf}}$       | -0.002                    | 0.009    | -0.162 | -1.5                            | 0.003                    |
| $R_{Vc}$                | 0.145                     | 0.240    | -0.110 | 9.8                             | 0.646                    |

In Table R1 we show that the difference between the comparisons with and without the selection criteria (AOD differences, the thresholds of fine/coarse mode AOD (AODf/c), Ångström exponent (AE) and AOD>0.02 at 440 nm instead of AOD>0.03 at 500 nm), is generally quite small. The selection criteria for AOD differences do not guarantee an improved comparison of the aerosol size properties (although it is expected and happened in several cases). For 5 out of 9 parameters the correlation factor shows minor increase for the 'unfiltered' dataset and for 4 parameters the median difference showed minor decrease or no change. Conflicting errors can cancel each other, certain AOD differences may not affect the inversions (e.g. due to similar AE), the 5-minute averages and inversion residual thresholds may have filtered out other outliers, while the limited numbers of remaining outliers do not affect the statistics significantly (Fig. R1). Several relative differences increase due to the

inclusion of lower aerosol loads. However, as we mentioned, the goal was not to keep the best results, but to show more clearly the differences originating from the combined uncertainties of the SD inversion method and the AOD retrieval methodology with as few effects from other factors as possible.

Figure R1: The AODf comparisons between GRASP-AOD from PFR data retrievals and AERONET inversions from radiance for the final pre-print dataset (a) filtered for AOD differences, inversion residuals, AOD>0.03, AODf>0.02 at 500 nm and AE>0.3 and the full dataset (b), filtered only for inversion residuals and AOD>0.02 at 440 nm. Due to the nature of the locations, a significant part of the dataset was filtered out with a small change in the minimum AOD threshold.

**Modifications**

We added 1 table same as table 1 and one same as table 2 for the 'unfiltered' data. We also added in the supplement the equivalent of figures 2-3 without selection criteria plus two additional panels in Fig. S1 corresponding to the 'unfiltered' data. We cite them in the related presentation of the results and discussion.

- -We removed the AE thresholds from the 'filtered' datasets and changed the minimum AOD thresholds to AOD>0.02 at 440 nm (all datasets) and  $AOD_{c/f}>0.01$  at 500 nm (concerns only the 'filtered' datasets).
- 1.238 Rephrased from:

'Finally, to ensure a better quality of comparisons that more clearly display the performance of GRASP, we filtered the datasets according to their AOD differences (PFR – CIMEL and AER-SKY – AER-SDA), AODf and AODc (AER-SKY – AER-SDA).'

To:

'Finally, to more clearly display the performance of GRASP-AOD, we include comparisons where we filtered the datasets according to their AOD differences. The aim of this procedure is to minimize the effect of factors such as cloud contamination and imperfect instrument synchronization under high variability conditions and better show the combined effect of uncertainties related to GRASP-AOD and the typical retrieval uncertainties of the AOD in the PFR observations (including factors such as uncertainties of calibration, gas absorption corrections, signal random noise and field of view).'

**b) BTS section**

**Explanation**

The same logic as in the previous response part can be applied here. However, in this case we had additional reasons to use a cleaning procedure through AOD comparisons:

- 1) Currently, the BTS dataset does not include any operational procedures for cloud screening and quality assurance as the instruments belonging to GAW-PFR and AERONET. It was not in the scope of this work to develop such procedures. However, the co-location of PFR/CIMEL and BTS provides the opportunity to use the PFR/CIMEL cloud screening algorithm and to extract a usable for our goal subset of synchronous measurements. In Gröbner et al. (2023) during a short-term intercomparison campaign, where the days were carefully selected, the BTS AOD showed good agreement with the reference. However, the 30 second synchronisation window used here introduces a minor possibility of cloud effects on AOD retrievals in occasions that sun visibility changes withing a few seconds.
- 2) The BTS AOD dataset comes from two spectroradiometers, but only one of the two includes directly comparable wavelengths with the PFR. Therefore, large errors due to technical issues with the second BTS cannot be identified directly by this comparison. Certain thresholds on the difference between the BTS observed AOD and the AOD extrapolated with a smooth function (e.g. Ångström approximation) can limit such cases. This criterion can be based on the results of Fig. 7, the AOD uncertainties and the results of the BTS-PFR AOD comparisons. As long as the criteria are not too strict, they remove unrealistic and highly suspicious data that do not severely reduce data availability and yet ensure sufficient quality to produce meaningful results with GRASP-AOD.

Below (Fig. R2 b) you can find the behaviour of data filtered only by the BTS-PFR comparison according to the thresholds lower in the response. AOD from 340 and 863 nm (same spectroradiometer) are well correlated. AOD from 863 nm and 1551 nm (different spectroradiometer) shows cases of AOD above 1 or 2 at 1551 nm, while AOD at 863 nm is below 0.05.

Figure R2: AOD scatter plots from BTS for two different pairs of wavelengths for the same data (filtered by comparing the BTS and the PFR).

**Modifications**

We modified the selection criteria to reduce the data affected by each large error source. Below are the sources considered and our strategy for each:

Error sources:

- 1. Malfunctions and other technical issues in the BTS measuring below 1000 nm, effect of thick clouds.
- 2. Remaining cloud contamination.
- 3. Malfunctions and other technical issues in the BTS measuring above 1000 nm.

Error minimisation strategies per source (details in table R3):

- 1. Reject data where the AOD differences are too large or AOD and AE are too high or too low.
- 2. From the data filtered by the previous condition reject all data with high AOD at 500 nm and low AE that do not satisfy the AOD difference conditions in preprint line 271.
- 3. From the data filtered by the previous conditions reject all data corresponding to AOD at 440 nm smaller than AOD at 1550 nm. Also, reject all data with large differences between observed and extrapolated AOD at wavelengths above 1000 nm.

Table R2: The conditions that the AOD of BTS dataset satisfies to be included in the final selection for the investigation and validation of size distribution parameter retrievals. The data are filtered by the values of AOD, their comparison with the PFR dataset and in case of wavelengths longer than 1000 nm, by comparison with the extrapolated AOD through the Ångström approximation.

| Condition number | Parameter value                                                                                                        | AOD difference
BTS-PFR                                                         | AOD difference of BTS observed-extrapolated                                                   |
|------------------|------------------------------------------------------------------------------------------------------------------------|-----------------------------------------------------------------------------------|-----------------------------------------------------------------------------------------------|
| Condition 1      | Retained only: • 0 <aod<2.5 0<ae<2.5<="" td="" •=""><td>ΔAOD<=0.1 (all common wavelengths)</td><td>-</td></aod<2.5> | ΔAOD<=0.1 (all common wavelengths)                                                | -                                                                                             |
| Condition 2      | Applicable when:  • AOD>0.3 at 500nm  • AE<0.5                                                                         | ΔAOD: • <=0.07 for 368.1 nm • <=0.05 for 412.1 and 500.7 nm • <=0.04 for 862.9 nm | -                                                                                             |
| Condition 3      | Retained only:
AOD 440 >AOD 1551                                                              | -                                                                                 | ΔAOD: • <0.025 at 1022 nm • <0.03 at 1238 nm • <0.035 1550 nm • <0.05 at 2108.1 and 2129.8 nm |

**Additional information and updated results:**

- AE and interpolated or extrapolated AOD are calculated with the methodology in section 2.4
- The thresholds for the applicability of condition 2 correspond approximately to the 90th percentile of AOD at 500 nm and 25th percentile of AE from the PFR data during the period September 2021-September 2024. AE<0.5 data are also flagged by both cloud screening methods as cloudy in Schenzinger and Kreuter (2021), Fig. 3. In Fig. R4, we show the comparisons with and without data filtering.
- The latest central wavelength to pixel assignment showed minor changes compared to the older dataset so we updated the wavelengths and AOD data. The updated wavelengths are: 340.2, 368.1, 380.3. 412.1, 440.3, 500.7, 675.4, 747.4, 780.6, 862.9, 870.0, 1022.0, 1238.0, 1551.0, 2108.1 and 2129.8 nm. The AOD differences with the previous version were mostly in the third or fourth decimal digit.
- We updated in the manuscript Fig. 7 according to the new data selection comparison and recalculation on the Ångström Exponent (found an underestimation of approximately 0.2 due to error in the previous processing). In this response, we include also statistics of the data filtered only by the BTS-PFR and the criterion AOD440>AOD1551 (Fig. R4). Fig. R5 shows the updated Fig. 7.
- We recalculated all SD parameters from GRASP using the new selection of wavelengths, AOD data and AE (affects the initial guesses).

- AOD difference thresholds in condition 3 were selected by taking into account the typical AOD uncertainty of PFR is 0.01, the BTS-PFR comparison (median of the unfiltered BTS-PFR differences at 500 nm is 0.002 and the 95th-5th percentile difference is 0.047), Fig. R4 and the low Davos AOD at the infrared wavelengths. We consider the thresholds not particularly restrictive (as shown below in allowing AOD features deviating from the Ångström approximation and accounting for the AOD uncertainties.
- We added in the manuscript section 3.2 the number of data and statistics of AOD differences (median, standard deviation and 95th-5th percentile difference) between BTS and PFR, with and without the selection criteria.

**Updated results and information about rejected data points:**

The initial common dataset (BTS-PFR) includes 28592 data points. Conditions (1) and (2) retained 93.4% of the initial data points. After applying the condition of AOD440>AOD1551 we retain 86.4% of the data. Fully applying condition (3) we retain 76.3% of the data points (21826). 41 measurements in that dataset showed negative AOD to 2108.1 nm, therefore the final dataset includes 21785.

Increasing the thresholds of the final condition to 0.04 for 1022, 1238, 1551 nm and 0.06 for 2108.1 and 2129.8 nm, would increase the data by 4.1% of the final dataset. We consider the final selection a sufficient large dataset with wide enough range of conditions (for example >7% of data correspond to AE<0.2 and 10% of data to AOD>0.2 at 500 nm), certain level of quality and AOD differences with the reference instrument close to the expected uncertainties. This dataset also satisfies already the condition of AOD at 440 nm >0.02. Of course, the dataset is not simulating the dataset of a lone BTS in an operational network, but it is suitable for the purpose of our work. Fig. R4 is the updated preprint Fig. 7 (final data selection).

Finally, applying the inversion residual quality criteria to both 7 and 16 wavelength selection output, we rejected  $\sim 0.3\%$  of the final AOD selected datapoints, to a final SD parameter retrievals of 21728 datapoints.

The validation of the re-processed GRASP output is in the response of major comment 2.

Figure R3: The differences of AOD between BTS and PFR for the whole dataset (a) and the final data selection (b) at 500 nm. The black lines show the general rejection threshold and the red lines show the rejection threshold at this wavelength under conditions suspected of cloud contamination. In panel (a) the 2.2% of the data points are out of the horizontal axis limits due to the maximum difference being 2.65 and the measurements per bin very few or zero, making it visually less comprehensive without smaller horizontal axis limits.

Figure R4: The difference statistics between the AOD observed by the BTS and AOD interpolated or extrapolated with the Ångström approximation before filtering the data according the differences between observed and fitted AOD.

Figure R5: The difference statistics between the AOD observed by the BTS and AOD interpolated or extrapolated with the Ångström approximation for the final dataset.

**2) Inversion residuals**

**Explanation**

Imposing such quality criteria to inversions is a typical practice when using data from AERONET (https://aeronet.gsfc.nasa.gov/new\_web/PDF/AERONETcriteria\_final1.pdf) and GRASP (Dubovik et al., 2011) to ensure a certain level of consistency between the forward model and inversion and the criteria are adapted for each case. The residual thresholds used in Torres and Fuertes (2021) were defined empirically for the version of CIMEL measuring up to 1020 nm. In this manuscript, we use the same criteria for the same 7 wavelengths. In the case of PFR, the wavelengths are fewer, so the model convergence is easier. Therefore, we used stricter residuals to achieve the same goal. In the case of BTS with 16 selected wavelengths, the criteria are in fact less strict than the ones in Torres and Fuertes (2021). The same happens with observed and extrapolated AOD, since the latter is smooth, but this is no longer relevant (response to major comment 2#). Our modified criteria retain the same logic and the dependence with the AOD. The value of the inversion residual threshold is never below 0.01, which is a typical AOD measurement uncertainty for air mass 1 (maximum uncertainty).

**Modifications**

- We added information about the number of data filtered through this procedure. The inversion residuals rejected 0.81% in Davos, 1.21% in Izana, 4.25% in Hohenpeissenberg and 5.45% in Lindenberg of the total GRASP-AOD inversions that satisfy the criterion AOD>0.02 at 440 nm.
- We modified the justification to be clearer.

**1.226 rephrased from:**

'To retrieve the SD parameters from PFR AOD, we used the multi-initial guesses approach described in Torres et al., (2017) and Torres and Fuertes, (2021) for the GRASP settings. We also used a modified version of the criteria in the same studies, to consider an inversion valid. To keep the inversions, the absolute inversion fitting error must be below 0.01 if the AOD at 412 nm is below 0.5 and below  $AOD_{412} \times 0.011 + 0.007$  if the AOD at 412 nm is above 0.5. The AOD absolute error at 500 nm has to be below  $0.01+0.005 \times AOD_{500}$ .'

To:

'To retrieve the SD parameters from PFR AOD, we used the multi-initial-guess approach described in Torres et al. (2017) and Torres & Fuertes (2021) for the GRASP settings. As proposed in those studies, we kept only data points corresponding to AOD > 0.02 at 440 nm. We also filtered the data according to the inversion residuals. A very high inversion residual indicates that the forward model failed to reproduce accurately the observed AOD provided as input, and therefore the output size distribution does not fit well that input.

Inversions using only the four PFR wavelengths make numerical convergence easier compared with CIMEL and lower residuals may appear without indicating a better-quality retrieval. Therefore, we used a modified version of the criteria mentioned in the two aforementioned studies to define a valid inversion. To retain an inversion, the absolute inversion fitting error must be below 0.01 if the AOD at 412 nm is below 0.5, and below  $AOD_{412} \times 0.011 + 0.007$  if the AOD at 412 nm is above 0.5. The AOD absolute error at 500 nm must be below  $0.01 + 0.005 \times AOD_{500}$ . These criteria are not particularly restrictive, as they always include a threshold that does not exceed 0.01 (the typical AOD uncertainty at air mass 1). These criteria rejected 0.81% of the total GRASP-AOD inversions in Davos, 1.21% in Izaña, 4.25% in Hohenpeissenberg, and 5.45% in Lindenberg, of those inversions that satisfied the criterion AOD > 0.02 at 440 nm.'

**3) Initial guesses**

**Explanation**

The GRASP initial guesses not only are necessary for the operation, but also affect the results. In the case of coarse mode volume median radius, GRASP-AOD with the particular spectral range provided the result is largely dictated by the initial guesses. This also affects other parameters. Therefore, the initial guess in that case has to be as 'well educated' as possible. The same applies to the refractive index assumption. In case an AERONET instrument is present it may seem redundant, but there are cases of common stations with several AERONET data gaps while at least one PFR was measuring (e.g. Davos and Marambio) plus the PFR schedule is more frequent. In most cases that an AERONET dataset is not available, other sources can be used such as LIDARs, in-situ measurements, satellite data or modelling. If no information is available or the reliability is considered too low, the initial guess of RVc and assumption of refractive index could be derived from nearby stations or others with similar characteristics. It was not in the scope of this study to perform a comparison of retrievals from all these potential sources of information. We just show what difference does it makes in the coarse mode characterisation the use of a single 'well educated' initial guess compared to the default selection of standard algorithm, under conditions of strong coarse mode presence. It's worth noting that the RVc bias did not disappear, which is evidence of the limitations of the retrieval methodology.

**Modifications**

Added justification.

Rephrased 1.465-466 from:

'Using prior knowledge for the GRASP-PFR retrievals -specifically a single initial guess for  $R_{\rm Ve}$ - may improve the results.'

To:

'Using prior information for the GRASP-PFR retrievals -specifically a single initial guess for  $R_{Vc}$ - may improve the results. This information (an  $R_{Vc}$  average or climatology) can be derived by the best available source in each location and time period, such as ground based

instruments that were measuring in that site for at least part of the period, satellite retrievals or modelling.'

**4) Selected conditions**

**Explanation**

The sub-section 3.1.3 is dedicated to the performance of GRASP-AOD under different conditions. The conditions in lines 366 and 546 are selected purely to display the difference in the comparisons, so to display the fact that these conditions are indeed favourable. This aligns with the purpose of that particular sub-section and the discussion related to it. These conditions are quite typical in the GAW-PFR network. Quality criteria such as AOD at 440 nm >0.2 or >0.4 are particularly restrictive due to the nature of the locations. Ångström exponent >1 on the other hand is not so restrictive as most of the GAW-PFR locations are affected mostly by fine mode aerosols for most of the time.

Regarding the criteria of  $AOD_{f/c}$  and Ångström exponent outside subsection 3.1.3 (lines 231-232), the idea was just to ensure that there was certain aerosol load of the corresponding mode in each case to make the size distribution parameters more meaningful. It is understandable that these thresholds may seem unnecessary at that part of the manuscript. However, in the case of BTS retrievals the coarse mode criterion remains meaningful. The focus of that section is mostly on  $R_{Vc}$ , but the location is usually under conditions dominated by fine mode and low aerosol loads. In that section (3.2) we include information about the results with and without this criterion already.

**Modifications**

-Rephrased 1. 360 from:

'In this section, we show that the radii can be improved by further restricting the datasets to more specific conditions.'

To:

'In this section, we investigate the performance of the radii in relation to the AOD and AE values and identify conditions under which the comparisons with AERONET inversions significantly improve.'

-We no longer use the criteria AODf>0.02 and AE>0.3 or AODc>0.02 and AE<1.8, but modified versions in specific cases as described in the first two parts of the response to this comment.

**5) Particularities in the case of Canadian wildfire smoke episode**

**Explanation**

The Canadian smoke episode was included in the analysis as it was an unusual and "extreme" case using either PFR or CIMEL as inputs. Due to the negative Angström exponent for

wavelengths lower than 500nm, we used this case as an example of investigating how GRASP could perform.

We showed that in that case, the low Ångström Exponent in fine mode dominated conditions, led to strong overestimation of the coarse mode under the settings used in earlier sections. Due to the rare occurrence of such conditions, we did not focus on modifications of the general algorithm settings to account for such events. One goal was to test whether the four channels of PFR are able to reproduce some of the features of AERONET inversions in such case with a different settings approach and how it compares with CIMEL. A second goal was to use the BTS to provide insights into the role of wavelength selection. In this case, the interest is stronger on the shortest wavelengths. More importantly, we wanted to see if the GRASP output can reproduce better the characteristics of this episode with different settings. As these settings do not impose an expectation of coarse mode domination based on Ångström Exponent low value and this led to a substantial improvement of the output, although the settings were not optimised for this particular episode.

**Modifications**

Added clearer justification.

-1.495: Rephrased from:

'To retrieve the SD from AOD, we did not use the Torres and Fuertes, (2021) settings approach, since the low AE in that episode led to settings more appropriate for dust cases. The resulting output included strong overestimation of  $C_{Vc}$  when  $AOD_c > 0.2$  at 500 nm and inversion residuals larger than our selection thresholds. Both AER-SKY and AER-SDA outputs showed  $AOD_c < 0.003$  during that smoke episode. Using single retrievals with more general settings (Supplement Sect. S5), we reproduced the aforementioned SD characteristics for both PFR and BTS AOD.'

To:

'To retrieve the SD from AOD, we did not use only the Torres and Fuertes, (2021) settings approach, since the low AE in that episode led to settings more appropriate for dust cases. The resulting output included strong overestimation of  $C_{Vc}$  when with  $AOD_c > 0.2$  at 500 nm and inversion residuals larger than our selection thresholds. Both AERONET inversions from almucantar scans and the direct sun spectral deconvolution algorithm outputs showed  $AOD_c < 0.003$  during that smoke episode. Therefore, we also tested single retrievals with more general settings (Supplement Sect. S5) to check if the large coarse mode overestimation was indeed due to the settings. Although the alternative settings are not optimized for this particular episode, we reproduced the aforementioned SD characteristics for both PFR and BTS AOD.'

- -1.500: Added 'as the most interesting wavelengths in this episode are the shorter ones.' before 'However'.
- 2. The focus of the study gets lost throughout the text: the title speaks about "Direct Solar Irradiance Measurements with High Temporal Resolution and Spectral Range". The abstract speaks about GAW-PFR, which provides AOD at 4 spectral channels. High temporal

resolution is true, but not high spectral range. And the rest of the manuscript includes a mixup of PFR, BTS, AERONET-SDA, CIMEL in various aspects (only direct, radiances). Is the main focus the application of GRASP-AOD to the GAW-PFR network (line 138); or something else? For instance, the use of BTS could me focused to assess what is lost when using the PFR spectral range instead of a larger one. In this way, it would contribute to the main goal of the study. Instead, it seems to be an attempt to explore the performance of the GRASP-AOD itself, which in my view is a different analysis.

Why/in which aspect is relevant for the study the high temporal resolution indicated in the title?

**By the way, the input for the GRASP-AOD retrieval is the AOD, not direct solar irradiance.**

In this study, the main idea was to continue the work of Torres and Fuertes (2021) by applying their method in a different network and instrument types, measuring only direct irradiance and investigate the performance in relation to the conditions.

The main questions are:

- How GRASP-AOD performs when applied to the PFR that includes only 4 wavelengths and a smaller spectral range than CIMEL?
- -How is the performance affected by the conditions (AOD, AE)?
- -The retrieval of the coarse mode median radius is challenging. Can a larger spectral range help?

Regarding the AERONET output parameters, we used them as 'reference' to validate PFR and BTS based retrievals or to derive information in order to test the GRASP-AOD retrievals under 'well educated' initial guesses. Further use is limited, parallel to PFR/BTS use and restricted to the Canadian wildfire event, which received 'special treatment' for reasons explained earlier in the response and in the manuscript.

**Below is a point-by-point response to the comment:**

- I. Title: The instruments measure direct solar irradiance and they use algorithms to retrieve AOD. To be more clear, we replaced 'direct solar irradiance' with 'aerosol optical depth' in the title of the manuscript.
- II. The high temporal resolution is relevant due to the use of instruments measuring every minute. This is a substantial increase from the AERONET inversions products that due to solar elevation related retrieval limitations are not available for a large part of the day. It is also the main motivation for the study and the GRASP-AOD application in general.
- III. Exactly as mentioned in the referee comment, we included the BTS (extended spectral range) with the main goal of understanding whether and to what extent it can provide some benefit for the retrieval of the coarse mode volume median radius. As shown, this parameter shows very little to no sensitivity to the AOD from the PFR. In Torres and Fuertes 2021, the finding was the same for only slightly longer spectral range (1020nm, which corresponds to

CIMEL standard version). The results showed an increase in sensitivity of the parameter to AOD and its correlation with AERONET. Due to data limitations our analysis is not conclusive for the accuracy and the effect of the conditions.

**Modifications:**

- -We modified the abstract according to the following:
- 1) 1.22: added '(known as the GRASP-AOD application)' after 'parameter'.
- 2) 1.24: Rephrased from:
- 'In this study, we selected four common stations of GAW-PFR and AERONET and used GRASP to retrieve the bimodal size distribution parameters from AOD measured by GAW-PFR instruments (PFRs). We assessed the homogeneity with the AERONET output parameters and investigated the effect of the spectral range and on such retrievals.'

To:

- 'In this study, we selected four common stations of GAW-PFR and AERONET, used GRASP to retrieve the bimodal size distribution parameters from AOD measured by GAW-PFR instruments (PFRs) and validated the results for different conditions using AERONET data as reference. One of those sites includes a multi-year parallel timeseries from two different BTS spectroradiometers that combined can provide direct spectral irradiance (and as a result AOD) in the 300-2150 nm range. Using this dataset we were able to investigate the effect and potential benefits of the increased spectral range and on GRASP-AOD retrievals. This is mostly focused on the retrieval of the coarse mode volume median radius, which is particularly challenging with the filter radiometers measuring up to 862 or 1020 nm.'
- -Section 2.4 now starts with: 'Torres and Fuertes (2021) and the previous parts of the study showed that the retrieval of  $R_{Vc}$  is particularly challenging with GRASP-AOD applied to PFR and CIMEL standard version data.'
- -Added ', especially for  $R_{Vc}$ ' in the end of the third sentence of section 2.4, directly after 'using GRASP'.
- -We modified and updated the results related to the effect of spectral range according to the following:
- 1) In Fig. 8 we removed the panels b and d. Panels a and b are updated according to the new data selection, their horizontal axes have the same limit and the text box includes percentile differences too ( $80^{th}$ - $20^{th}$ ). The bin width is now the same for both graphs.
- 2) Tables 4 and 5 will no longer include AOD-ext. They are replaced by Tables R3-R4.
- 3) We removed all mentions to GRASP-AOD retrievals from AOD interpolated or extrapolated.

Table R3: Statistics of the differences between the GRASP-AOD retrievals from BTS data when using 16 wavelengths minus when using 7 wavelengths. The first row corresponds to AOD at 500 nm simulated by GRASP for each one of the two wavelength selections.

| Parameter | median | St.dev. | P95th-P5th | R | Number | of |  |
|-----------|--------|---------|------------|---|--------|----|--|
|-----------|--------|---------|------------|---|--------|----|--|

|                    | difference |       |       |      | measurements |
|--------------------|------------|-------|-------|------|--------------|
| AOD fitted         | 0.000      | 0.001 | 0.003 | 1.00 | 21728        |
| $AOD_{\mathrm{f}}$ | -0.005     | 0.008 | 0.019 | 0.98 | 21728        |
| $AOD_c$            | 0.005      | 0.008 | 0.018 | 0.99 | 21728        |
| $C_{VT}$           | 0.005      | 0.017 | 0.048 | 0.95 | 21728        |
| $C_{\mathrm{Vf}}$  | 0.000      | 0.002 | 0.005 | 0.94 | 21728        |
| $C_{Vc}$           | 0.006      | 0.017 | 0.047 | 0.95 | 21728        |
| $R_{\text{eff}}$   | 0.047      | 0.198 | 0.661 | 0.80 | 21728        |
| $R_{\mathrm{Vf}}$  | -0.026     | 0.032 | 0.100 | 0.81 | 21728        |
| $R_{\mathrm{Vc}}$  | 0.029      | 0.364 | 1.189 | 0.10 | 21728        |

Table R4: Statistics of the differences between the GRASP-AOD from BTS data retrievals and the AERONET inversions from almucantar scans for sixteen wavelengths in the 340-2130 nm range and for seven wavelengths in the 340-1022 nm range. The first row of data shows the comparison of the AOD at 500 nm corresponding to the AOD BTS observations and the almucantar scan inversions.

|                           | 16 waveleng          | ths     |      | 7 wavelengtl         | 18      |       |                        |    |
|---------------------------|----------------------|---------|------|----------------------|---------|-------|------------------------|----|
| Parameter                 | median
difference | St.dev. | R    | median
difference | St.dev. | R     | Number
measurements | of |
| AOD obs.                  | 0.003                | 0.009   | 0.99 | 0.003                | 0.009   | 0.99  | 63                     |    |
| $\mathrm{AOD}_\mathrm{f}$ | -0.010               | 0.010   | 0.87 | -0.002               | 0.012   | 0.84  | 63                     |    |
| $AOD_c$                   | 0.010                | 0.011   | 0.99 | 0.003                | 0.010   | 0.99  | 63                     |    |
| $C_{VT}$                  | 0.002                | 0.022   | 0.96 | 0.003                | 0.018   | 0.94  | 63                     |    |
| $C_{\mathrm{Vf}}$         | -0.003               | 0.005   | 0.59 | -0.003               | 0.005   | 0.59  | 63                     |    |
| $C_{Vc}$                  | 0.007                | 0.020   | 0.96 | 0.004                | 0.017   | 0.94  | 63                     |    |
| $R_{\text{eff}}$          | 0.139                | 0.136   | 0.78 | 0.210                | 0.240   | 0.85  | 63                     |    |
| $R_{\mathrm{Vf}}$         | 0.008                | 0.051   | 0.15 | 0.030                | 0.067   | 0.29  | 63                     |    |
| $R_{\mathrm{Vc}}$         | -0.469               | 0.428   | 0.73 | -0.387               | 0.626   | -0.04 | 63                     |    |

**3) Table 6 is replaced by table R5.**

Table R5: Statistics of the differences between the GRASP-AOD from BTS data retrievals and the AERONET inversions from almucantar scans for sixteen wavelengths in the 340- 2130 nm range and for seven wavelengths in the 340-1022 nm range focused on the coarse mode. The data correspond to  $AOD_c > 0.02$  and FMF < 0.8 at 500 nm.

|           | 16 wavelengths       |         |      | 7 wavelengtl         | 18      |      |                        |    |
|-----------|----------------------|---------|------|----------------------|---------|------|------------------------|----|
| Parameter | median
difference | St.dev. | R    | median
difference | St.dev. | R    | Number
measurements | of |
| $AOD_c$   | 0.008                | 0.014   | 0.98 | -0.002               | 0.010   | 0.98 | 30                     |    |
| $C_{Vc}$  | -0.003               | 0.024   | 0.93 | 0.008                | 0.023   | 0.90 | 30                     |    |

| R eff  | 0.169  | 0.150 | 0.60 | 0.398 | 0.186 | 0.62 | 30 |
|-------------------|--------|-------|------|-------|-------|------|----|
| $R_{\mathrm{Vc}}$ | -0.275 | 0.390 | 0.64 | 0.253 | 0.484 | 0.16 | 30 |

- IV. We applied the modifications to discussion related to spectral range effect to reduce the size, make it more focused to the main purpose and removed the discussion related to the Ångström approximation. The modifications are as follows:
- Deleted the following parts:
- 1) 1. 565: 'four different'
- 2) 1. 568: 'For both wavelength selections we used the observed AOD and the AOD estimated through Eq. (2).'
- 3) 1.578: 'We also found no consistent improvement to all properties when using one particular AOD dataset over the retrievals from the other three.'
- 4) 1.614: 'The comparison between AER-SKY and GRASP-BTS in most cases shows smaller differences when using the observed AOD compared to the extrapolated with the same wavelength selection, despite the noise of the observed AOD. This is an indication that cases where reality deviates significantly from the Ångström law may result in observable effects on the SD parameter retrievals. Therefore, the observed AOD or a more representative smooth function should be used for this purpose, but still the limited data does not allow high confidence or generalized conclusions.'
- 5) l. 622: 'Using the AOD-obs dataset at seven wavelengths from BTS, we found the best consistency between the GRASP-PFR and GRASP-BTS comparisons against AER-SKY, which was expected'
- -l. 565: Added 'potential improvements in the retrieval of  $R_{Vc}$  with larger spectral range and generally' after 'To assess'.
- 1.568: Rephrased from:

'Our results showed that the wavelength selection affects the results regardless of the use of the observed AOD or a smooth spectral AOD function corresponding to the Ångström law. The magnitude of the differences varied depending on the compared datasets and the parameter under study. Regarding the wavelength selections, we found that using the larger spectral range,  $R_{\rm Vc}$  is no longer stuck very close to the initial guess (as it happens for the PFR and CIMEL spectral ranges).'

To:

'Our results showed that the larger spectral range had minor effect for most parameters, but  $R_{Vc}$  retrievals are no longer mostly stuck very close to the initial guess (as it happens for the PFR and CIMEL spectral ranges).'

-Updated the numbers in the rest of the discussion about the spectral range effects according to the results in the above tables.

- -l. 580 added after 'was': 'the substantial increase of the correlation with AERONET for  $R_{Vc}$  and'. Changed 'effect' to 'effects'.
- -1.668: added before 'our': 'Therefore, the larger spectral range increases the algorithm's capabilities for  $R_{Vc}$  retrieval, which is the main limitation of the smaller spectral range and the same time seems to preserve the performance in the other parameters.'

**Other comments:**

Line18: "they can be every minute" The verb is missing. Please rewrite.

'be' is the verb.

Rephrased from:

'AOD measurements are significantly more frequent as they can be even every minute and are affected only by clouds being too close or covering the solar disk.'

To:

'AOD is measured significantly more frequently as there can be one measurement even every minute. Also, the AOD measurements are affected only by clouds being too close or covering the solar disk.'

*L23: Size >> Particle size.*

**Corrected.**

L36: "GRASP-BTS AOD-based dataset": this is hard to understand. There are plenty of such acronyms in the manuscript. The GRASP papers (https://www.grasp-open.com/publications-2/) frequently adopt a nick for frequent applications: GRASP-AOD is one of them (Torres and Fuertes, 2021). And that's the application (settings included) that is used in this case. Therefore the name cannot be changed depending on the instrument (GRASP-BTS, GRASP-PFR, and so on). This creates much of confusion. Many other instances in the manuscript (e.g. line 297) are in the same direction.

Modified to 'GRASP-AOD from PFR data' and 'GRASP-AOD from BTS data'.

Introduction

It is overall too long in providing aerosol generalities that are well known and could be much reduced by citing a few key references. Better come to the point, focus in introducing the elements that are needed in this particular study.

We deleted several sentences from the introduction.

List of removed sentences and words or phrases:

1. 1. 51 'The influence of aerosols on solar radiation serves as a key driver for climate and weather patterns (IPCC, 2023).'

- 2.1.54 'particularly those with radii under  $2.5 \mu m$ , which are major contributors to premature mortality,'
- 3. 1. 58 ' It is mathematically represented through the Beer-Lambert-Bouguer law:

$$I = I_o e^{-m\tau} \tag{1}$$

where I is the solar irradiance at the surface, Io is the irradiance at the top of the atmosphere, m represents the air mass coefficient and  $\tau$  the atmospheric optical depth. The optical depth is the sum of the optical depth from all atmospheric components, so AOD is a component of  $\tau$ . 'Eq. 2 will become Eq. 1.

In line 64 the sentence now will be 'AOD spectral dependence can be also approximated by the Ångström law'

- 4. 1. 70 'Sun photometers are the primary tools for AOD measurements, measuring DSI at specific wavelengths.'
- 5. 1. 80 'In this study, we focus on GAW-PFR and AERONET.'
- 6. 1. 86 'or single scattering albedo (SSA)'
- 7. 1. 91 'the SSA'
- 8. 1. 108 'different'
- 9. l. 109 '(Ezhova et al., 2028)'
- 10. 1. 110 ', including relative response between different wavelengths (Pandolfi et al., 2018). Size is particularly important for the computation of the aerosol asymmetry factor (Andrews et al., 2006; Ehlers and Moosmüller, 2023) as the asymmetry factor and phase function show significant sensitivity to size (Li et al., 2022).'
- 11. l. 117 'Monteiro et al., 2018; Shao et al., 2020; Konsta et al., 2021;'
- 12. 1. 118 'However, volcanic aerosol sizes vary significantly depending on the type, so they can contribute to a larger extent in the fine mode. A volcanic eruption may either increase or decrease the aerosol size locally (Martin et al., 2008; Wrana et al., 2023).'
- 13. l. 121 'or even with a significant contribution of ultra fine particles through combustion for industrial, heating and transport purposes (Tiwari et al., 2014; Zhang et al., 2022; Abdillah et al., 2024). Coarse mode particles are also emitted though, mostly through mechanical processes (Wu and Boor, 2021).'
- 14. l. 128 'This can also lead to implications in modelling cloud properties, such as droplet number concentration and cloud albedo, depending on the aerosol size distribution used (Kodros and Pierce, 2017) and the radiative forcing attribution to aerosols and clouds (Virtanen et al., 2025). Reduced cloud coverage also seems to be the main reason for the unusually high global temperature in 2023 that was not solely explained by anthropogenic global warming due to greenhouse gas emissions and the El Niño-Southern Oscillation phase (Goessling et al., 2024), where the role of aerosols remains yet unclear.'
- 15. 1.134 'As larger particles tend to be more massive, their residence time in the atmosphere is decreased due to gravity.'

The previous sentence will now be: 'Rodríguez-Arias et al., 2023), with larger particles showing reduced residence time in the atmosphere.'

16. l. 135 'are also responsible for various health effects and their size is one of the key parameters to describe those effects. Depending on the'

and

1.137 'they can infiltrate and affect different parts of the body'

The sentence will now continue as: 'size is also related to health effects with smaller aerosols being typically more dangerous (Kodros et al., 2018).'

*L65: correlates with aerosol COLUMN concentration.*

**Corrected.**

L66: size >> size predominance. This must be changed throughout the text (e.g. Line 113). Otherwise it would speak about a single size instead of a size distribution in which a specific size range dominates (in terms of mass or volume, by the way; not number).

**Corrected.**

L90-94: the AERONET inversion uses AOD and sky radiances (not only radiances, not only almucantar geometry). The version 3 inversions (Sinyuk et al) could be cited here.

**Rephrased from:**

'are typically retrieved through the inverse modelling of sky radiance observations at the almucantar geometry (Dubovik and King, 2000) observations at the almucantar geometry.'

To:

'are typically retrieved through the inverse modelling of sky radiance observations and AOD (Dubovik and King, 2000; Sinyuk et al., 2020).'

L105: it may be worth mentioning here that index of refraction needs to be known or assumed.

Added 'and prior knowledge or assumption of the aerosol refractive index' before '(Torres et al., 2017)'.

L111: this is very well known since decades. A classic reference to some old book is more desirable here; otherwise it looks like a recent finding. Same for line 123.

Added in the references for both cases the book: 'An Introduction to Atmospheric Radiation Second Edition' by K. N. Liou, 2002, 1980, Elsevier Science (USA), ISBN: 0-12-451451-0.

Section 2

**Line 184: the 1640nm channel is missing.**

It is not available for all stations and all years used in this study (and generally). That is what we mean in that line with 'at least'. But we mention 1640 nm in the previous line.

L188: 30 seconds >> 1 minute.

**Corrected.**

L190: the alignment is done by the 4 quadrant detector, not the collimator. The latter is used for straylight rejection in the sky scans.

**Rephrased from:**

'A silicon detector records the radiation, while the instrument's 1.2° FOV ensures precise solar alignment. To further enhance accuracy, a four-quadrant detector identifies the point of maximum solar intensity, ensuring the instrument points directly at the Sun'

To:

'A silicon detector records the radiation, while the instrument's 1.2° FOV to reject stray light during the sky radiance scans. To further enhance accuracy, a four-quadrant detector identifies the point of maximum solar intensity, ensuring the instrument points directly at the Sun'

L201: Sun angular size in 0.5deg and FOV is 3deg. It's unavoidable that some sky light leaks into the DSI.

Indeed, although this effect should be limited in our case. We included information about the AOD comparison between BTS and PFR to provide information about the validity of the AOD. In general, this effect is minimal for small particles and evident mostly under strong presence of coarse mode aerosols such as dust and sea salt, but also smaller wavelengths (Takamura & Irie, 2019; Zhao et al., 2012).

For instance, in Cuevas et al., 2019 there is investigation on the effect of FoV difference between CIMEL (1.3°) and PFR (2.5°). It can explain a significant portion of AOD differences between CIMEL and PFR, outside the WMO limits for AOD>0.1 in Izana (dusty conditions). This percentage is 53% of the total differences outside the limits for 380 nm and reduces to 13% for 500 nm. For AOD<0.3 and particle radius below 1.5 μm, their results showed that the effect on AOD at 380 nm was less than 0.01. An increase of FoV to 3° from 2.5° in AOD<0.1 with significant or dominant fine mode component the effect should be even smaller.

L204: please provide a reference on the BTS calibration. Same about the AOD retrieval with BTS (line 209).

Added 'Gröbner et al., 2023'in both cases.

L213: not all those inputs are needed for GRASP to run. Please rewrite.

All mentioned parameters at this point are written as the output parameters of GRASP except the AOD, which is written as the input. Of course, all output parameters need a corresponding initial guess to be provided as input either.

**Rephrased from**

'AOD at more than one wavelength provides retrievals of the SD parameters ( $C_{Vf}$ ,  $C_{Vc}$ ,  $R_{Vf}$ ,  $R_{Vc}$ ,  $\sigma_{Vf}$  and  $\sigma_{Vc}$ ) as a main output'

To:

'AOD at more than one wavelength combined with an assumption of the refractive index, provides as the main output retrievals of the SD parameters ( $C_{Vf}$ ,  $C_{Vc}$ ,  $R_{Vf}$ ,  $R_{Vc}$ ,  $\sigma_{Vf}$  and  $\sigma_{Vc}$ )'

L214: the GRASP-AOD requires AOD plus the assumption of index of refraction.

Corrected according to the response on the previous comment.

L216: check repetition.

Removed 'Using the SD parameters, we can also compute CVT and Reff'.

*L221: exclusively >> separately?.*

Replaced with 'separately'.

*L226: PFR AOD >> AOD obtained with the PFR (to improve readability).*

Corrected.

Line 230: AOD500nm>0.03. Why? What do GRASP-AOD papers say about this minimum AOD?

We replaced the criterion with the one in Torres and Fuertes (2021): AOD440 nm>0.02.

Same about the criteria on Angstrom exponent and fine/coarse mode AOD. It brings me again to the main comment #1.

Response in main comment #1.

L250: Why 1.45-0.003? Some justification is needed. Otherwise it looks totally arbitrary. Same for line 288.

This value is selected as typical for continental European and urban sites (applicable for 3 out of 4 stations in our case), based on climatologies of Dubovik et al 2002. Theoretically this would not apply to Izana, but the Izana climatology also did not show very large deviation.

We included this explanation and reference in the sentence.

L258: the 1640nm channel (available in the Cimel) is not used in this list. Why?.

In 1640 nm the gas absorptions are stronger than 1551 nm, where the AOD could be useable even without correction (Gröbner et al., 2023). Since the 2 wavelengths are not that far from each other and the spectral range extends beyond 2000 nm, this channel was not necessary for the purpose of that section.

This channel is also not available to all AERONET instruments, particularly during earlier years and did not show substantial effect in other studies (Torres et al., 2017; Torres and Fuertes, 2021).

Figure 1: I would recommend changing to Table format.

Changed to table format and removed the AOD-ext related filters.

Section 3

L297: AER-SKY: not only radiances are used (as already mentioned). And the abbreviations are awful.

This did not imply that only radiances are used. It was just a way to separate from SDA output since both are AERONET products. Response below to the reply of the comment about table 1.

L300: "In this section": check repetitions.

Replaced 'In this section,' with 'here'.

L303: better quantify than give the valuation ("Excellent"). The goodness will depend on the intended application or the uncertainty estimates.

Replaced 'excellent' with the median differences and standard deviations.

Table 1: table captions are preferably self-explicative, but this is not possible with all the abbreviations..

Changed to 'AERONET radiance inversions', 'AERONET direct sun AOD' and 'AERONET SDA output'.

Replaced the last sentence with: 'Both comparisons include only the selected data according to the criteria in section S2'.

L345: the differences between using climatology or fix value are small. What about using the closest AERONET retrieval of the refractive index? This magnitude is very much related to aerosol composition and can depart significantly from the site average value.

The data availability of the refractive index is already very limited by the frequency of almucantar scans and cloud screening. It becomes much more limited in the level 2 quality assured dataset that require high enough AOD, which is scarce in these locations. The use of level 1.5 would require certain procedures to reduce the uncertainty that would also limit further the available data, while it would include significant percentage of measurements with larger uncertainties than the level 2. Therefore, it cannot be applicable or an improvement for most of the measurements.

Climatology is a piece of information that can be available in a location that certain instrumentation (such as a CIMEL sun and sky photometer) was used for several years or through alternative methodologies and can be applied to the entire dataset. The idea of that sub-section was to investigate what happens if the climatology is not available, but we can use one fixed value that is expected to be close to the typical ones for those locations, in comparison to the case that AERONET climatology is available. Therefore, we expect that the refractive differences should not be that large if the fixed value is indeed sensible for what is expected for the site.

*L362: depend >> affect.*

Corrected.

*L381:* extrapolated >> interpolated?.

Some wavelengths are within the range of the ones used in the Ångström law and others extend beyond this range. Replaced with 'interpolated or extrapolated'.

L383: reference needed.

Added 'Eck et al., 2023'.

L399: please clarify why is this expected.

These 2 datasets differ both in spectral range and AOD estimation method. The other pairs of datasets have at least one in common, so they are closer to each other in terms of AOD provided as input to GRASP. However, this is no longer relevant as we removed the results and discussion related to the AOD-ext datasets.

L466-467: you try to retrieve the SD, therefore you don't know it. What does it mean "using prior knowledge"? Moreover, the use of AERONET inversion information as initial guess to improve the result is a trick (main comment #1). The approach in line 564 is far more reasonable.

The response to this comment is in the response of the main comment #1.

Figure 11: excellent example of data not fitting to the Angstrom law. Why is the fit to the Angstrom law used in the manuscript? Is it justified? Why not using just AOD observations? This is actually realized by the authors (line 618).

A brief explanation of the use of the Ångström law is in lines 262-265. The aim was not to imply that this AOD estimation is equally or more accurate or representative compared to the observed AOD. The AOD interpolated and extrapolated from Ångström law was added for two reasons:

- 1) To test how much GRASP inversions change from the use of those different wavelength selections in cases of no AOD noise or any features and information content in the longer wavelengths that can affect the results. Just to test the GRASP response on that.
- 2) To examine the need to use observations of these additional wavelengths. If the Ångström law was an accurate and representative AOD interpolation or extrapolation under all circumstances, it could be used on PFR or CIMEL data to extend the spectral range, in case this is helpful for the retrieval algorithm. The fact that this is not the case, shows the need for actual observations in those wavelengths in order to improve retrievals of aerosol properties.

We removed the related parts except Figure 7 (explained in the response of the major comment #2).

L500: not that unusual. See Eck et al (1999; 2023).

We did not claim that it never happened before (we even cited already Eck et al., 2023 and now added a new related reference, Masoom et al., 2025). However, the previous occurrences do not mean this case is not unusual, less interesting or less deserving of the additional work related to it.

L572: is 1640nm channel considered in the Cimel? It's almost 2 times longer than the 862nm of the PFR..

Added 'standard version' after 'CIMEL'.

L590: please re-think. It's obvious that the lesser number of data, the better fit. With 2 points a linear fit is just perfect. But there must be more information (lower uncertainty) if more data are used in the input. Moreover, using the Angstrom approximation makes you lose real AOD features (see my previous comment on that).

The explanation in lines 584 to 590 does not refer to the Ångström approximation datasets. It refers to the comparison of the median differences between AERONET retrievals and retrievals from BTS AOD for the two wavelength selections under the data selection in the table. The median RVc difference between GRASP-AOD using BTS AOD and AERONET is smaller for the shorter spectral range wavelength selection in both cases of AOD datasets (observed or Ångström approximation) for this particular selection of data (which contains very few measurements though).

Using smaller spectral range leads the  $R_{Vc}$  GRASP inversion not to deviate significantly from the initial guess in most cases, because  $R_{Vc}$  shows very low sensitivity to AOD in shorter wavelengths. This is not a result of better or worse fit, it is a lower 'capability' to perform the retrieval through the inversion model. It is also related to the Lagrange parameter of initial guess (Herrera et al., 2022), which constraints the  $R_{Vc}$ , since its impact is larger for fewer wavelengths. When we use  $R_{Vc}$  initial guess close to the average of AERONET and 1022 nm as the longest wavelength, the  $R_{Vc}$  output will compare well with AERONET  $R_{Vc}$  if the latter does not show large variability from the average. Using wavelengths longer than 2000 nm created some sensitivity of  $R_{Vc}$  to AOD, but this does not ensure low enough retrieval uncertainty to provide more accurate retrievals compared to the aforementioned case of shorter spectral range under all circumstances.

The smaller  $R_{Vc}$  variability under shorter spectral range leads to a small random uncertainty component. So, the accuracy is dictated by the accuracy of the initial guess. In the case of a longer spectral range, the variability of  $R_{Vc}$  increased as we can see in Figure 8, regardless of the AOD dataset. The correlation factor also increased, which suggests that part of this increased variability is related to reality, but part of it should still be related to the retrieval uncertainties.

Table A1:

GRASP-BTS: ...and GRASP-AOD as retrieval.

Corrected according to the related responses in earlier comments.

AER-SKY: the input is AOD and radiance (almucantar or hybrid geometries).

Corrected according to the related responses in earlier comments.

**Modifications not requested by the referee:**

1) Added a citation regarding the Canadian wildfires in line 482 (Section 3.3.2).

Masoom, A., Kazadzis, S., Modini, R. L., Gysel-Beer, M., Gröbner, J., Coen, M. C., Navas-Guzman, F., Kouremeti, N., Brem, B. T., Nowak, N. K., Martucci, G., Hervo, M., and Erb, S.: Long range transport of Canadian Wildfire smoke to Europe in Fall 2023: aerosol properties and spectral features of smoke particles, EGUsphere [preprint], https://doi.org/10.5194/egusphere-2025-2755, 2025.

**Rephrased in lines 482-483:**

'Canadian wildfires caused unusual AOD observations (Masoom et al., 2025), where the highest AOD occurred at 500 nm rather than the shortest available wavelength, leading to'

To:

- 'Canadian wildfires caused unusual AOD observations in several locations (Masoom et al., 2025) including Davos, where the highest AOD occurred at 500 nm rather than the shortest available wavelength. This led to'
- 2) Line 608: changes 'retrained' to 'retained'.
- 3) Added acknowledgements about the local operators for the PFR and CIMEL datasets.

**References cited in the response:**

Cuevas, E., Romero-Campos, P. M., Kouremeti, N., Kazadzis, S., Räisänen, P., García, R. D., Barreto, A., Guirado-Fuentes, C., Ramos, R., Toledano, C., Almansa, F., and Gröbner, J.: Aerosol optical depth comparison between GAW-PFR and AERONET-Cimel radiometers from long-term (2005–2015) 1 min synchronous measurements, Atmos. Meas. Tech., 12, 4309–4337, https://doi.org/10.5194/amt-12-4309-2019, 2019.

Dubovik, O., Holben, B., Eck., T. F., Smirnov, A., Kaufman, Y. J., King, M. D., Tanre, D., and Slutzker, I.: Variability of absorption and optical properties of key aerosol types observed in worldwide locations, J. Atmos. Sci., 59, 590–608, 2002.

Dubovik, O., Herman, M., Holdak, A., Lapyonok, T., Tanré, D., Deuzé, J. L., Ducos, F., Sinyuk, A., and Lopatin, A.: Statistically optimized inversion algorithm for enhanced retrieval of aerosol properties from spectral multi-angle polarimetric satellite observations, Atmos. Meas. Tech., 4, 975–1018, https://doi.org/10.5194/amt-4-975-2011, 2011.

Herrera, M. E., Dubovik, O., Torres, B., Lapyonok, T., Fuertes, D., Lopatin, A., Litvinov, P., Chen, C., Benavent-Oltra, J. A., Bali, J. L., and Ristori, P. R.: Estimates of remote sensing retrieval errors by the GRASP algorithm: application to ground-based observations, concept and validation, Atmos. Meas. Tech., 15, 6075–6126, https://doi.org/10.5194/amt-15-6075-2022, 2022.

Gröbner, J., Kouremeti, N., Hülsen, G., Zuber, R., Ribnitzky, M., Nevas, S., Sperfeld, P., Schwind, K., Schneider, P., Kazadzis, S., Barreto, Á., Gardiner, T., Mottungan, K., Medland, D., and Coleman, M.: Spectral aerosol optical depth from SI-traceable spectral solar irradiance measurements, Atmos. Meas. Tech., 16, 4667–4680, https://doi.org/10.5194/amt-16-4667-2023, 2023.

Schenzinger, V. and Kreuter, A.: Reducing cloud contamination in aerosol optical depth (AOD) measurements, Atmos. Meas. Tech., 14, 2787–2798, https://doi.org/10.5194/amt-14-2787-2021, 2021.

Takamura, T., & Irie, H.: Forward Scattering Effect on the Estimation of the Aerosol Optical Thickness for Sun Photometry. Journal of the Meteorological Society of Japan. Ser. II, 97, 6, 1211-1219, 2019.

Torres, B. and Fuertes, D.: Characterization of aerosol size properties from measurements of spectral optical depth: a global validation of the GRASP-AOD code using long-term AERONET data, Atmos. Meas. Tech., 14, 4471–4506, https://doi.org/10.5194/amt-14-4471-2021, 2021.

Torres, B., Dubovik, O., Fuertes, D., Schuster, G., Cachorro, V. E., Lapyonok, T., Goloub, P., Blarel, L., Barreto, A., Mallet, M., Toledano, C., and Tanré, D.: Advanced characterisation of aerosol size properties from measurements of spectral optical depth using the GRASP algorithm, Atmos. Meas. Tech., 10, 3743–3781, https://doi.org/10.5194/amt-10-3743-2017, 2017.

Zhao, F., Tan, Y., Li, Z., and Gai, C.: The effect and correction of aerosol forward scattering on retrieval of aerosol optical depth from Sun photometer measurements, Geophysical Research Letters, 39, 14, https://doi.org/10.1029/2012GL052135, 2012.

Table R6: List of abbreviations.

| GAW-PFR | Global Atmospheric Watch-Precision Filter Radiometer       |
|---------|------------------------------------------------------------|
| AERONET | Aerosol Robotic Network                                    |
| GRASP   | Generalized Retrieval of Atmosphere and Surface Properties |
| PFR     | Precision Filter Radiometer                                |
| CIMEL   | CIMEL CE318-TS sun and sky photometer                      |
| BTS     | The array spectroradiometer 'BiTec Sensor'.                |
| POM     | PREDE-POM sun and sky radiometer                           |
| AOD     | Aerosol Optical Depth                                      |
| AE      | Ångström Exponent                                          |
|         |                                                            |

AODf Fine Mode Aerosol Optical Depth

AODc Coarse Mode Aerosol Optical Depth

CvT Total Volume Concentration

Cvf Fine Mode Volume Concentration

Cvc Coarse Mode Volume Concentration

Reff Effective Radius

**R**Vf Fine Mode Volume Median Radius

RVc Coarse Mode Volume Median Radius

**FMF** Fine mode fraction of AOD

**SD** Aerosol Size Distribution

FoV Field-of-View Angle

St.dev. Standard Deviation

**P95** 95th percentile

P5 5th percentile

**R** Pearson correlation factor

RMSE Root mean square error

**AOD-obs** AOD retrieved directly from the direct spectral irradiance measured by an instrument.

AOD-ext AOD estimated by the Ångström law after calculation of the Ångström exponent and

turbidity coefficient using observed spectral AOD.